# Deep Layers as Stochastic Solvers

**Adel Bibi** [*]
KAUST

**Bernard Ghanem**
KAUST

**Vladlen Koltun**
Intel Labs

**René Ranftl**
Intel Labs

## Abstract

We provide a novel perspective on the forward pass through a block of layers in a deep network. In particular, we show that a forward pass through a standard dropout layer followed by a linear layer and a non-linear activation is equivalent to optimizing a convex objective with a single iteration of a $\tau$-nice Proximal Stochastic Gradient method. We further show that replacing standard Bernoulli dropout with additive dropout is equivalent to optimizing the same convex objective with a variance-reduced proximal method. By expressing both fully-connected and convolutional layers as special cases of a high-order tensor product, we unify the underlying convex optimization problem in the tensor setting and derive a formula for the Lipschitz constant $L$ used to determine the optimal step size of the above proximal methods. We conduct experiments with standard convolutional networks applied to the CIFAR-10 and CIFAR-100 datasets and show that replacing a block of layers with multiple iterations of the corresponding solver, with step size set via $L$, consistently improves classification accuracy.

## 1 Introduction

Deep learning has revolutionized computer vision and natural language processing and is increasingly applied throughout science and engineering (LeCun et al., 2015). This has motivated the mathematical analysis of various aspects of deep networks, such as the capacity and uniqueness of their representations (Soatto & Chiuso, 2014; Papyan et al., 2018) and their global training convergence properties (Haeffele & Vidal, 2017). However, a complete characterization of deep networks remains elusive. For example, Bernoulli dropout layers are known to improve generalization (Srivastava et al., 2014), but a thorough theoretical understanding of their behavior remains an open problem. While basic dropout layers have proven to be effective, there are many other types of dropout with various desirable properties (Molchanov et al., 2017). This raises many questions. Can the fundamental block of layers that consists of a dropout layer followed by a linear transformation and a non-linear activation be further improved for better generalization? Can the choice of dropout layer be made independently from the linear transformation and non-linear activation? Are there systematic ways to propose new types of dropout?

We attempt to address some of these questions by establishing a strong connection between the forward pass through a block of layers in a deep network and the solution of convex optimization problems of the following form:

$$\underset{\mathbf{x}\in\mathbb{R}^d}{\text{minimize}}\ F(\mathbf{x}) + g(\mathbf{x}), \quad F(\mathbf{x}) \stackrel{\text{def}}{=} \frac{1}{n}\sum_{i}^{n} f_i(\mathbf{a}_i^\top \mathbf{x}). \tag{1}$$

Note that when $f_i(\mathbf{a}_i^\top\mathbf{x}) = \frac{1}{2}(\mathbf{a}_i^\top\mathbf{x} - \mathbf{y}_i)^2$ and $g(\mathbf{x}) = \|\mathbf{x}\|_2^2$, Eq. (1) is standard ridge regression. When $g(\mathbf{x}) = \|\mathbf{x}\|_1$, Eq. (1) has the form of LASSO regression.

We show that a block of layers that consists of dropout followed by a linear transformation (fully-connected or convolutional) and a non-linear activation has close connections to applying stochastic solvers to (1). Interestingly, the choice of the stochastic optimization algorithm gives rise to commonly used dropout layers, such as Bernoulli and additive dropout, and to a family of other types of dropout layers that have not been explored before. As a special case, when the block in question does not include dropout, the stochastic algorithm reduces to a deterministic one.

Our contributions can be summarized as follows. (**i**) We show that a forward pass through a block that consists of Bernoulli dropout followed by a linear transformation and a non-linear activation

---

[*]The work was done during an internship at Intel Labs.

is equivalent to a single iteration of $\tau$-nice Proximal Stochastic Gradient, Prox-SG (Xiao & Zhang, 2014) when it is applied to an instance of (1). We provide various conditions on $g$ that recover (either exactly or approximately) common non-linearities used in practice. (**ii**) We show that the same block with an additive dropout instead of Bernoulli dropout is equivalent to a single iteration of mS2GD (Konečný et al., 2016) – a mini-batching form of variance-reduced SGD (Johnson & Zhang, 2013) – applied to an instance of (1). (**iii**) By expressing both fully-connected and convolutional layers (referred to as linear throughout) as special cases of a high-order tensor product (Bibi & Ghanem, 2017), we derive a formula for the Lipschitz constant $L$ of $\nabla F(\mathbf{x})$. As a consequence, we can compute the optimal step size for the stochastic solvers that correspond to blocks of layers. We note that concurrent work (Sedghi et al., 2019) used a different analysis strategy to derive an equivalent result for computing the singular values of convolutional layers. (**iv**) We validate our theoretical analysis experimentally by replacing blocks of layers in standard image classification networks with corresponding solvers and show that this improves the accuracy of the models.

## 2 RELATED WORK

Optimization algorithms can provide insight and guidance in the design of deep network architectures (Vogel & Pock, 2017; Kobler et al., 2017; Yang et al., 2016; Zhang & Ghanem, 2018). For example, Yang et al. (2016) have proposed a deep network architecture for compressed sensing. Their network, dubbed ADMM-Net, is inspired by ADMM updates (Boyd et al., 2011) on the compressed sensing objective. Similarly, Zhang & Ghanem (2018) demonstrated that unrolling a proximal gradient descent solver (Beck & Teboulle, 2009) on the same problem can further improve performance. The work of Kobler et al. (2017) demonstrated a relation between incremental proximal methods and ResNet blocks; based on this observation, they proposed a new architecture (variational networks) for the task of image reconstruction. Amos & Kolter (2017) proposed to embed optimization problems, in particular linearly-constrained quadratic programs, as structured layers in deep networks. Meinhardt et al. (2017) replaced proximal operators in optimization algorithms by neural networks. Huang & Van Gool (2017) proposed a new matrix layer, dubbed ReEig, that applies a thresholding operation to the eigenvalues of intermediate feature representations that are stacked in matrix form. ReEig can be tightly connected to a proximal operator of the set of positive semi-definite matrices. Sulam et al. (2018) proposed a new architecture based on a sparse representation construct, Multi-Layer Convolutional Sparse Coding (ML-CSC), initially introduced by Papyan et al. (2017). Sparsity on the intermediate representations was enforced by a multi-layer form of basis pursuit.

This body of work has demonstrated the merits of connecting the design of deep networks with optimization algorithms in the form of structured layers. Yet, with few exceptions (Amos & Kolter, 2017; Sulam et al., 2018), previous works propose specialized architectures for specific tasks. Our work aims to contribute to a unified framework that relates optimization algorithms to deep layers.

A line of work aims to provide rigorous interpretation for dropout layers. For example, Wager et al. (2013) showed that dropout is linked to an adaptively balanced $\ell_2$-regularized loss. Wang & Manning (2013) showed that approximating the loss with a normal distribution leads to a faster form of dropout. Gal & Ghahramani (2016a;b) developed a framework that connects dropout with approximate variational inference in Bayesian models. We provide a complementary perspective, in which dropout layers arise naturally in an optimization-driven framework for network design.

## 3 UNIFIED FRAMEWORK

This section is organized as follows. We introduce our notation and preliminaries in Section 3.1. In Section 3.2, we present a motivational example relating a single iteration of proximal gradient descent (Prox-GD) on (1) to the forward pass through a fully-connected layer followed by a non-linear activation. We will show that several commonly used non-linear activations can be exactly or approximately represented as proximal operators of $g(\mathbf{x})$. In Section 3.3, we unify fully-connected and convolutional layers as special cases of a high-order tensor product. We propose a generic instance of (1) in a tensor setting, where we provide a formula for the Lipschitz constant $L$ of the finite sum structure of (1). In Section 3.4, we derive an intimate relation between stochastic solvers, namely $\tau$-nice Prox-SG and mS2GD, and two types of dropout layers. Figure 1 shows an overview of the connections that will be developed.

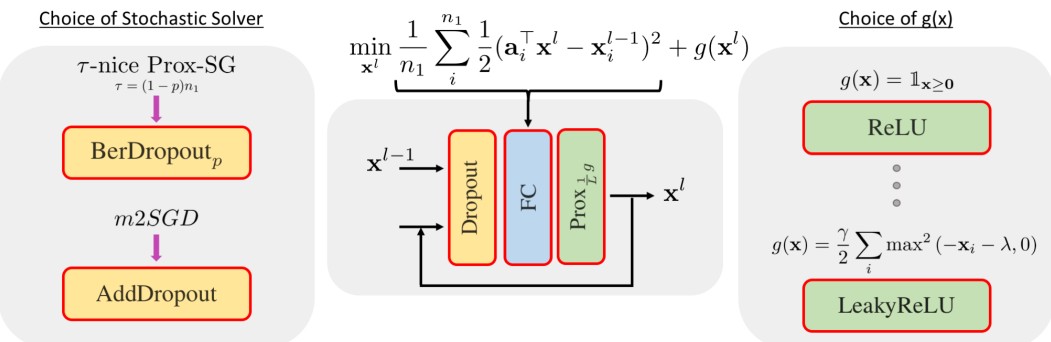

Figure 1: An overview of the tight relation between a single iteration of a stochastic solver and the forward pass through the $l^{\text{th}}$ layer in a network that consists of dropout followed by a linear transformation and a non-linear activation. We study an instance of problem (1) with quadratic $F(\mathbf{x})$, where $\mathbf{x}^{l-1}$ are the input activations and $\mathbf{x}^l$, the variables being optimized, correspond to the output activations. Varying the type of stochastic solver changes the nature of the dropout layer, while the prior $g(\mathbf{x})$ on the output activations determines the non-linearity $\text{Prox}_{\frac{1}{L}g}(.)$.

## 3.1 NOTATION AND PRELIMINARIES

As we will be working with tensors, we will follow the tensor notation of Kolda & Bader (2009). The order of a tensor is the number of its dimensions. In particular, scalars are tensors of order zero, vectors are tensors of order one, and matrices are tensors of order two. We denote scalars by lowercase letters $a$, vectors by bold lowercase letters $\mathbf{a}$, and matrices by bold capital letters $\mathbf{A}$. We use subscripts $\mathbf{a}_i$ to refer to individual elements in a vector. Tensors of order three or more will be denoted by cursive capital letters $\mathcal{A} \in \mathbb{R}^{J_1 \times J_2 \times \cdots \times J_n}$. Throughout the paper, we will handle tensors that are of at most order four. High-order tensors with a second dimension of size equal to one are traditionally called vector tensors and denoted $\vec{\mathcal{A}} \in \mathbb{R}^{J_1 \times 1 \times J_3 \times J_4}$. We use $\mathcal{A}(i, j, k, z)$ to refer to an element in a tensor and $\mathcal{A}(i, j, k, :)$ to refer to a slice of a tensor. The inner product between tensors of the same size is denoted $\langle \mathcal{A}, \mathcal{B} \rangle = \sum_{i_1, \ldots, i_N} \mathcal{A}(i_1, \ldots, i_N) \mathcal{B}(i_1, \ldots, i_N)$. The squared Frobenius norm of a tensor $\mathcal{A}$ is defined as $\|\mathcal{A}\|_F^2 = \langle \mathcal{A}, \mathcal{A} \rangle$. Lastly, the superscripts $\top$ and $\mathbf{H}$ are used to denote the transpose and the Hermitian transpose, respectively.

## 3.2 MOTIVATIONAL INSIGHT: NON-LINEAR ACTIVATIONS AS PROXIMAL OPERATORS

As a motivating example, we consider the $l^{\text{th}}$ linear layer in a deep network that is followed by a non-linear activation $\rho$, i.e. $\mathbf{x}^l = \rho(\mathbf{A}\mathbf{x}^{l-1} + \mathbf{b})$, where $\mathbf{A} \in \mathbb{R}^{n_2 \times n_1}$ and $\mathbf{b} \in \mathbb{R}^{n_2}$ are the weights and biases of the layer and $\mathbf{x}^{l-1}$ and $\mathbf{x}^l$ are the input and output activations, respectively. Now consider an instance of (1) with a convex function $g(\mathbf{x})$ and

$$F(\mathbf{x}^l) = \frac{1}{2}\|\mathbf{A}^\top \mathbf{x}^l - \mathbf{x}^{l-1}\|^2 - \mathbf{b}^\top \mathbf{x}^l = \frac{1}{2}\sum_i^{n_1}(\mathbf{A}^\top(i,:)\mathbf{x}^l - \mathbf{x}_i^{l-1})^2 - \mathbf{b}^\top \mathbf{x}^l, \qquad (2)$$

where $\mathbf{A}^\top(i,:)$ is the $i^{\text{th}}$ row of $\mathbf{A}^\top$. Such an objective can be optimized iteratively in $\mathbf{x}^l$ using Prox-GD with the following update equation:

$$\mathbf{x}^l \leftarrow \text{Prox}_{\frac{1}{L}g}\left(\left(\mathbf{I} - \frac{1}{L}\mathbf{A}\mathbf{A}^\top\right)\mathbf{x}^l + \frac{1}{L}\left(\mathbf{A}\mathbf{x}^{l-1} + \mathbf{b}\right)\right), \qquad (3)$$

where the Lipschitz constant $L = \lambda_{\max}\left(\mathbf{A}\mathbf{A}^\top\right)$ and $\lambda_{\max}(.)$ denotes the maximum eigenvalue.

By initializing the iterative optimization at $\mathbf{x}^l = \mathbf{0}$, it becomes clear that a single iteration of (3) is equivalent to a fully-connected layer followed by a non-linearity that is implemented by the proximal operator (Fawzi et al., 2015). The choice of $g(\mathbf{x})$ determines the specific form of the non-linearity $\rho$. Several popular activation functions can be traced back to their corresponding $g(\mathbf{x})$. The ReLU, which enforces non-negative output activations, corresponds to the indicator function $g(\mathbf{x}) = \mathbb{1}_{\mathbf{x} \geq 0}$; the corresponding instance of problem (1) is a non-negative quadratic program. Similar observations for the ReLU have been made in other contexts (Amos & Kolter, 2017; Papyan et al., 2017). We

| $g(\mathbf{x})$ | $\mathbb{1}_{\mathbf{x}\geq 0}$ | $\frac{\gamma}{2}\sum_i \max^2\left(-\mathbf{x}_i - \lambda, 0\right)$ | $-\gamma\sum_i \log(\mathbf{x}_i)$ | $-\gamma\sum_i \log(1 - \mathbf{x}_i^2)$ |
|---|---|---|---|---|
| $\mathrm{Prox}_g\left(\eta\right)$ | $\max(0, \eta)$ | $\begin{cases} \frac{\eta - \gamma\lambda}{1+\gamma} & \text{if } \eta \leq -\lambda \\ \eta & \text{if } \eta \geq -\lambda \end{cases}$ | $\frac{1}{2}\eta + \sqrt{\frac{1}{4}\eta^2 + \gamma}$ | Root of cubic polynomial |
| Shape | | | | |
| Activation | $= \mathrm{ReLU}(\eta)$ | $= \mathrm{LeakyReLU}(\eta)$ | $\approx \mathrm{Softplus}(\eta)$ | $\approx \mathrm{Tanh}(\eta)$ |

Table 1: Different choices of $g(\mathbf{x})$, their corresponding proximal operators, and their relation to common activation functions. Squared hinge loss regularization of the activations yields a generalized Leaky ReLU. Log-barriers recover smooth activations, such as SoftPlus, Tanh, or Sigmoid. Derivations can be found in supplementary material.

observe that many other activation functions fit this framework. For example, when $g(\mathbf{x})$ is a squared hinge loss, i.e. $\frac{\gamma}{2}\sum_i \max^2\left(-\mathbf{x}_i - \lambda, 0\right)$, a single update of (3) is equivalent to a linear layer followed by a Leaky ReLU. Table 1 lists some other choices of $g(\mathbf{x})$ and their induced activations.

Note that $g(\mathbf{x})$ is not required to exhibit a simple, coordinate-wise separable structure. More complex functions can be used, as long as the proximal operator is easy to evaluate. Interesting examples arise when the output activations have matrix structure. For instance, one can impose nuclear norm regularization $g(\mathbf{X}) = \|\mathbf{X}\|_*$ to encourage $\mathbf{X}$ to be low rank. Alternatively, one can enforce positive semi-definite structure on the matrix $\mathbf{X}$ by defining $g(\mathbf{X}) = \mathbb{1}_{\mathbf{X}\succeq\mathbf{0}}$. A similar activation has been used for higher-order pooling (Huang & Van Gool, 2017).

In what follows, we will show that this connection can be further extended to explain dropout layers. Interestingly, specific forms of dropout do not arise from particular forms of objective (1), but from different stochastic optimization algorithms that are applied to it.

### 3.3 UNIFYING FULLY-CONNECTED AND CONVOLUTIONAL LAYERS

Before presenting our main results on the equivalence between a forward pass through a block of layers and solving (1) with stochastic algorithms, we provide some key lemmas. These lemmas will be necessary for a unified treatment of fully-connected and convolutional layers as generic linear layers. This generic treatment will enable efficient computation of the Lipschitz constant for both fully-connected and convolutional layers.

**Lemma 1.** Consider the $l^{\text{th}}$ convolutional layer in a deep network with some non-linear activation, e.g. $\mathrm{Prox}_g(.)$, where the weights $\mathcal{A} \in \mathbb{R}^{n_2 \times n_1 \times W \times H}$, biases $\vec{\mathcal{B}} \in \mathbb{R}^{n_2 \times 1 \times W \times H}$, and input activations $\vec{\mathcal{X}}^{l-1} \in \mathbb{R}^{n_1 \times 1 \times W \times H}$ are stacked into $4^{\text{th}}$-order tensors. We can describe the layer as

$$\vec{\mathcal{X}}^l = \mathrm{Prox}_g\left(\mathcal{A} \circledast_{\mathrm{HO}} \vec{\mathcal{X}}^{l-1} + \vec{\mathcal{B}}\right), \tag{4}$$

where $\circledast_{\mathrm{HO}}$ is the high-order tensor product. Here $n_1$ is the number of input features, $n_2$ is the number of output features (number of filters), and $W$ and $H$ are the spatial dimensions of the features. As a special case, a fully-connected layer follows naturally, since $\circledast_{\mathrm{HO}}$ reduces to a matrix-vector multiplication when $W = H = 1$.

The proof can be found in supplementary material. Note that the order of the dimensions is essential in this notation, as the first dimension in $\mathcal{A}$ corresponds to the number of independent filters while the second corresponds to the input features that will be aggregated after the 2D convolutions. Also note that according to the definition of $\circledast_{\mathrm{HO}}$ in (Bibi & Ghanem, 2017), the spatial size of the filters in $\mathcal{A}$, namely $W$ and $H$, has to match the spatial dimensions of the input activations $\vec{\mathcal{X}}^{l-1}$, since the operator $\circledast_{\mathrm{HO}}$ performs 2D circular convolutions while convolutions in deep networks are 2D linear convolutions. This is not a restriction, since one can perform linear convolution through a zero-padded circular convolution. Lastly, we assume that the values in $\vec{\mathcal{B}}$ are replicated along the spatial dimensions $W$ and $H$ in order to recover the behaviour of biases in deep networks.

Given this notation, we will refer to either a fully-connected or a convolutional layer as a linear layer throughout the rest of the paper. Since we are interested in a generic linear layer followed by

a non-linearity, we will consider the tensor quadratic version of $F(\mathbf{x})$, denoted $F(\vec{\mathcal{X}})$:

$$\underset{\vec{\mathcal{X}}}{\arg\min} \frac{1}{n_1} \sum_i^{n_1} \underbrace{\frac{n_1}{2} \|\mathcal{A}^{\mathbf{H}}(i,:,:,:) \circledast_{\text{HO}} \vec{\mathcal{X}} - \vec{\mathcal{X}}^{l-1}\|_F^2 - \langle \vec{\mathcal{B}}, \vec{\mathcal{X}} \rangle}_{f_i(\vec{\mathcal{X}})} + g(\vec{\mathcal{X}}). \tag{5}$$

Note that if $\mathcal{A} \in \mathbb{R}^{n_2 \times n_1 \times W \times H}$, then $\mathcal{A}^{\mathbf{H}} \in \mathbb{R}^{n_1 \times n_2 \times W \times H}$, where each of the frontal slices of $\mathcal{A}(:,:,i,j)$ is transposed and each filter, $\mathcal{A}(i,j,:,:)$, is rotated by $180°$. This means that $\mathcal{A}^{\mathbf{H}} \circledast_{\text{HO}} \vec{\mathcal{X}}$ aggregates the $n_2$ filters after performing 2D correlations. This is performed $n_1$ times independently. This operation is commonly referred to as a transposed convolution. Details can be found in supplementary material.

Next, the following lemma provides a practical formula for the computation of the Lipschitz constant $L$ of the finite sum part of (5):

**Lemma 2.** The Lipschitz constant $L$ of $\nabla F(\vec{\mathcal{X}})$ as defined in (5) is given by

$$L = \max_{i \in \{1,2,...,W\}, j \in \{1,2,...,H\}} \left\{ \lambda_{\max} \left( \hat{\mathcal{A}}(:,:,i,j) \, \hat{\mathcal{A}}^{\mathbf{H}}(:,:,i,j) \right) \right\}, \tag{6}$$

where $\hat{\mathcal{A}}$ is the $2D$ discrete Fourier transform along the spatial dimensions $W$ and $H$.

The proof can be found in supplementary material. Lemma 2 states that the Lipschitz constant $L$ is the maximum among the set of maximum eigenvalues of all the possible $W \times H$ combinations of the outer product of frontal slices $\hat{\mathcal{A}}(:,:,i,j) \hat{\mathcal{A}}^{\mathbf{H}}(:,:,i,j)$. Note that if $W = H = 1$, then $\hat{\mathcal{A}} = \mathcal{A} \in \mathbb{R}^{n_2 \times n_1}$ since the 2D discrete Fourier transform of scalars (i.e. matrices of size $1 \times 1$) is an identity mapping. As a consequence, we can simplify (6) to $L = \max_{i=j=1}\{\lambda_{\max}(\mathcal{A}(:,:,i,j)\mathcal{A}^{\mathbf{H}}(:,:,i,j))\} = \lambda_{\max}(\mathcal{A}\mathcal{A}^{\top})$, which recovers the Lipschitz constant for fully-connected layers.

## 3.4 Dropout Layers as Variants of Stochastic Solvers

In this subsection, we present two propositions. The first shows the relation between standard Bernoulli dropout ($p$ is the dropout rate), $\text{BerDropout}_p$ (Srivastava et al., 2014), and $\tau$-nice Prox-SG. The second proposition relates additive dropout, AddDropout, to mS2GD (Konečný et al., 2016). We will first introduce a generic notion of sampling from a set. This is essential as the stochastic algorithms sample unbiased function estimates from the set of $n_1$ functions in (5).

**Definition 3.1.** (Gower et al., 2018). A sampling is a random set-valued mapping with values being the subsets of $[n_1] = \{1, \ldots, n_1\}$. A sampling $S$ is $\tau$-nice if it is uniform, i.e. $\text{Prob}(i \in S) = \text{Prob}(j \in S) \; \forall \, i, j$, and assigns equal probabilities to all subsets of $[n_1]$ of cardinality $\tau$ and zero probability to all others.

Various other types of sampling can be found in (Gower et al., 2018). We are now ready to present our first proposition.

**Proposition 1.** A single iteration of Prox-SG with $\tau$-nice sampling $S$ on (5) with $\tau = (1-p)n_1$, zero initialization, and unit step size can be shown to exhibit the update

$$\text{Prox}_{\frac{1}{L} g} \left( \frac{n_1}{\tau} \sum_{i \in S} \mathcal{A}(:,i,:,:) \circledast_{HO} \vec{\mathcal{X}}^{l-1}(i,:,:,:) + \vec{\mathcal{B}} \right), \tag{7}$$

which is equivalent to a forward pass through a $\text{BerDropout}_p$ layer that drops exactly $n_1 p$ input activations followed by a linear layer and a non-linear activation.

We provide a simplified sketch for fully-connected layers here. The detailed proof is in the supplement. To see how (7) reduces to the functional form of $\text{BerDropout}_p$ followed by a fully-connected layer and a non-linear activation, consider $W = H = 1$. The argument of $\text{Prox}_{\frac{1}{L} g}$ in (7) (without the bias term) reduces to

$$\frac{n_1}{\tau} \sum_{i \in S} \mathcal{A}(:,i,:,:) \circledast_{HO} \vec{\mathcal{X}}^{l-1}(i,:,:,:) = \frac{n_1}{\tau} \sum_{i \in S} \mathcal{A}(:,i) \vec{\mathcal{X}}^{l-1}(i) = \frac{n_1}{\tau} \mathcal{A} \; \text{BerDropout}_p \left( \vec{\mathcal{X}}^{l-1} \right). \tag{8}$$

The first equality follows from the definition of $\circledast_{HO}$, while the second equality follows from trivially reparameterizing the sum, with $\text{BerDropout}_p(.)$ being equivalent to a mask that zeroes out exactly $pn_1$ input activations. Note that if $\tau$-nice Prox-SG was replaced with Prox-GD, i.e. $\tau = n_1$, then this corresponds to having a $\text{BerDropout}_p$ layer with dropout rate $p = 0$; thus, (8) reduces to $\mathcal{A} \ \text{BerDropout}_p(\vec{\mathcal{X}}^{l-1}) = \mathcal{A}\vec{\mathcal{X}}^{l-1}$, which recovers our motivating example (3) that relates Prox-GD with the forward pass through a fully-connected layer followed by a non-linearity. Note that Proposition 1 directly suggests how to apply dropout to convolutional layers. Specifically, complete input features from $n_1$ should be dropped and the 2D convolutions should be performed only on the $\tau$-sampled subset, where $\tau = (1 - p)n_1$.

Similarly, the following proposition shows that a form of additive dropout, AddDropout, can be recovered from a different choice of stochastic solver.

**Proposition 2.** A single outer-loop iteration of mS2GD (Konečný et al., 2016) with unit step size and zero initialization is equivalent to a forward pass through an AddDropout layer followed by a linear layer and a non-linear activation.

The proof is given in the supplement. It is similar to Proposition 1, with mS2GD replacing $\tau$-nice Prox-SG. Note that any variance-reduced algorithm where one full gradient is computed at least once can be used here as a replacement for mS2GD. For instance, one can show that the serial sampling version of mS2GD, S2GD (Konečný et al., 2016), and SVRG (Johnson & Zhang, 2013) can also be used. Other algorithms such as Stochastic Coordinate Descent (Richtárik & Takáč, 2016) with arbitrary sampling are discussed in the supplement.

## 4 EXPERIMENTS

A natural question arises as a consequence of our framework: If common layers in deep networks can be understood as a single iteration of an optimization algorithm, what happens if the algorithm is applied for multiple iterations? We empirically answer this question in our experiments. In particular, we embed solvers as a replacement to their corresponding blocks of layers and show that this improves the accuracy of the models without an increase in the number of network parameters.

**Experimental setup.** We perform experiments on CIFAR-10 and CIFAR-100 (Krizhevsky & Hinton, 2009). In all experiments, training was conducted on $90\%$ of the training set while $10\%$ was left for validation. The networks used in the experiments are variants of LeNet (LeCun et al., 1999), AlexNet (Krizhevsky et al., 2012), and VGG16 (Simonyan & Zisserman, 2014). We used stochastic gradient descent with a momentum of $0.9$ and a weight decay of $5 \times 10^{-4}$. The learning rate was set to $(10^{-2}, 10^{-3}, 10^{-4})$ for the first, second, and third 100 epochs, respectively. For finetuning, the learning rate was initially set to $10^{-3}$ and reduced to $10^{-4}$ after 100 epochs. Moreover, when a block of layers is replaced with a deterministic solver, i.e. Prox-GD, the step size is set to the optimal constant $1/L$, where $L$ is computed according to Lemma 2 and updated every epoch without any zero padding as a circular convolution operator approximates a linear convolution in large dimensions (Zhu & Wakin, 2017). In Prox-SG, a decaying step size is necessary for convergence; therefore, the step size is exponentially decayed as suggested by Bottou (2012), where the initial step size is again set according to Lemma 2. Finally, to guarantee convergence of the stochastic solvers, we add the strongly convex function $\frac{\lambda}{2}\|\vec{\mathcal{X}}\|_F^2$ to the finite sum in (5), where we set $\lambda = 10^{-3}$ in all experiments. Note that for networks that include a stochastic solver, the network will be stochastic at test time. We thus report the average accuracy and standard deviation over 20 trials.

**Replacing fully-connected layers with solvers.** In this experiment, we demonstrate that (**i**) training networks with solvers replacing one or more blocks of layers can improve accuracy when trained from scratch, and (**ii**) the improvement is consistently present when one or more blocks are replaced with solvers at different layers in the network. To do so, we train a variant of LeNet on the CIFAR-10 dataset with two $\text{BerDropout}_p$ layers. The last two layers are fully-connected layers with ReLU activation. We consider three variants of this network: Both fully-connected layers are augmented with $\text{BerDropout}_p$ (LeNet-D-D), only the last layer is augmented with $\text{BerDropout}_p$ (LeNet-ND-D), and finally only the penultimate layer is augmented with $\text{BerDropout}_p$ (LeNet-D-ND). In all cases, we set the dropout rate to $p = 0.5$. We replace the $\text{BerDropout}_p$ layers with their corresponding stochastic solvers and run them for 10 iterations with $\tau = n_1/2$ (the setting corresponding to a dropout rate of $p = 0.5$). We train these networks from scratch using the same procedure as the baseline networks.

The results are summarized in Table 2. It can be seen that replacing BerDropout$_p$ with the corresponding stochastic solver ($\tau$-nice Prox-SG) improves performance significantly, for any choice of layer. The results indicate that networks that incorporate stochastic solvers can be trained stably and achieve desirable generalization performance.

|  | LeNet-D-D | LeNet-D-ND | LeNet-ND-D |
|---|---|---|---|
| Baseline | 64.39% | 71.72% | 68.54% |
| Prox-SG | **72.86% ± 0.177** | **75.20% ± 0.205** | **76.23% ± 0.206** |

Table 2: Comparison in accuracy between variants of the LeNet architecture on the CIFAR-10 dataset. The variants differ in the location (D or ND) and number of BerDropout$_p$ layers for both the baseline networks and their stochastic solver counterpart Prox-SG. Accuracy consistently improves when Prox-SG is used. Accuracy is reported on the test set.

**Convolutional layers and larger networks.** We now demonstrate that solvers can be used to improve larger networks. We conduct experiments with variants of AlexNet[1] and VGG16 on both CIFAR-10 and CIFAR-100. We start by training strong baselines for both AlexNet and VGG16, achieving 77.3% and 92.56% test accuracy on CIFAR-10, respectively. Note that performance on this dataset is nearly saturated. We then replace the first convolutional layer in AlexNet with the deterministic Prox-GD solver, since this layer is not preceded by a dropout layer. The results are summarized in Table 3. We observe that finetuning the baseline network with the solver leads to an improvement of $\approx 1.2\%$, without any change in the network's capacity. A similar improvement is observed on the harder CIFAR-100 dataset.

|  | AlexNet | AlexNet-Prox-GD |
|---|---|---|
| CIFAR-10 | 77.30% | **78.51%** |
| CIFAR-100 | 44.20% | **45.53%** |

Table 3: Replacing the first convolutional layer of AlexNet by the deterministic Prox-GD solver yields consistent improvement in test accuracy on CIFAR-10 and CIFAR-100.

Results on VGG16 are summarized in Table 4. Note that VGG16 has two fully-connected layers, which are preceded by a BerDropout$_p$ layer with dropout rate $p = 0.5$. We start by replacing only the last layer with Prox-SG with 30 iterations and $\tau = n_1/2$ (VGG16-Prox-SG-ND-D). We further replace both fully-connected layers that include BerDropout$_p$ with solvers (VGG16-Prox-SG-D-D). We observe comparable performance for both settings on CIFAR-10. We conjecture that this might be due to the dataset being close to saturation. On CIFAR-100, a more pronounced increase in accuracy is observed, where VGG-16-Prox-SG-ND-D outperforms the baseline by about 0.7%.

We further replace the stochastic solver with a deterministic solver and leave the dropout layers unchanged. We denote this setting as VGG16-Prox-GD in Table 4. Interestingly, this setting performs the best on CIFAR-10 and comparably to VGG16-Prox-SG-ND-D on CIFAR-100.

|  | VGG16 | VGG16-Prox-SG-ND-D | VGG16-Prox-SG-D-D | VGG16-Prox-GD |
|---|---|---|---|---|
| CIFAR-10 | 92.56% | 92.44% ± 0.028 | 92.57% ± 0.029 | **92.80%** |
| CIFAR-100 | 70.27% | 70.95% ± 0.042 | 70.44% ± 0.077 | **71.10%** |

Table 4: Experiments with the VGG16 architecture on CIFAR-10 and CIFAR-100. Accuracy is reported on the test set.

**Dropout rate vs. $\tau$-nice sampling.** In this experiment, we demonstrate that the improvement in performance is still consistently present across varying dropout rates. Since Proposition 1 has established a tight connection between the dropout rate $p$ and the sampling rate $\tau$ in (5), we observe that for different choices of dropout rate the baseline performance improves upon replacing a block of layers with a stochastic solver with the corresponding sampling rate $\tau$. We conduct experiments with

---

[1]AlexNet (Krizhevsky et al., 2012) was adapted to account for the difference in spatial size of the images in CIFAR-10 and ImageNet (Deng et al., 2009). The first convolutional layer has a padding of 5, and all max-pooling layers have a kernel size of 2. A single fully-connected layer follows at the end.

VGG16 on CIFAR-100. We train four different baseline models with varying choices of dropout rate $p \in \{0, 0.1, 0.9.0.95\}$ for the last layer. We then replace this block with a stochastic solver with a sampling rate $\tau$ and finetune the network.

Table 5 reports the accuracy of the baselines for varying dropout rates $p$ and compares to the accuracy of the stochastic solver with corresponding $\tau$ (Prox-SG). With a high dropout rate, the performance of the baseline network drops drastically. When using the stochastic solver, we observe a much more graceful degradation. For example, with a sampling rate $\tau$ that corresponds to an extreme dropout rate of $p = 0.95$ (i.e. 95% of all input activations are masked out), the baseline network with BerDropout$_p$ suffers a 56% reduction in accuracy while the stochastic solver declines by only 5%.

| Baseline | | Prox-SG | |
|---|---|---|---|
| Dropout rate $p$ | Accuracy | Sampling rate $\tau$ | Accuracy |
| 0 | 70.57% | 512 | **70.87** |
| 0.10 | **70.56**% | 461 | 70.51% $\pm$ 0.0198 |
| 0.50 | 70.27% | 256 | **70.95% $\pm$ 0.0419** |
| 0.90 | 68.34% | 51 | **69.19% $\pm$ 0.0589** |
| 0.95 | 30.61% | 26 | **67.42% $\pm$ 0.0774** |

Table 5: Comparison of the VGG16 architecture trained on CIFAR-100 with varying dropout rates $p$ in the last BerDropout$_p$ layer. We compare the baseline to its stochastic solver counterpart with corresponding sampling rate $\tau = (1 - p)n_1$. Accuracy is reported on the test set.

In summary, our experiments show that replacing common layers in deep networks with stochastic solvers can lead to better performance without increasing the number of parameters in the network. The resulting networks are stable to train and exhibit high accuracy in cases where standard dropout is problematic, such as high dropout rates.

## 5 DISCUSSION

We have presented equivalences between layers in deep networks and stochastic solvers, and have shown that this can be leveraged to improve accuracy. The presented relationships open many doors for future work. For instance, our framework shows an intimate relation between a dropout layer and the sampling $S$ from the set $[n_1]$ in a stochastic algorithm. As a consequence, one can borrow theory from the stochastic optimization literature to propose new types of dropout layers. For example, consider a serial importance sampling strategy with Prox-SG to solve (5) (Zhao & Zhang, 2015; Xiao & Zhang, 2014), where serial sampling is the sampling that satisfies Prob $(i \in S, j \in S) = 0$. A serial importance sampling $S$ from the set of functions $f_i(\vec{\mathcal{X}})$ is the sampling such that Prob $(i \in S) \propto \|\nabla f_i(\vec{\mathcal{X}})\| \propto L_i$, where $L_i$ is the Lipschitz constant of $\nabla f_i(\vec{\mathcal{X}})$, i.e. each function from the set $[n_1]$ is sampled with a probability proportional to the norm of the gradient of the function. This sampling strategy is the optimal serial sampling $S$ that maximizes the rate of convergence solving (5) (Zhao & Zhang, 2015). From a deep layer perspective, performing Prox-SG with importance sampling for a single iteration is equivalent to a forward pass through the same block of layers with a new dropout layer. Such a dropout layer will keep each input activation with a non-uniform probability proportional to the norm of the gradient. This is in contrast to BerDropout$_p$ where all input activations are kept with an equal probability $1 - p$. Other types of dropout arise when considering non-serial importance sampling where $|S| = \tau > 1$.

In summary, we have presented equivalences between stochastic solvers on a particular class of convex optimization problems and a forward pass through a dropout layer followed by a linear layer and a non-linear activation. Inspired by these equivalences, we have demonstrated empirically on multiple datasets and network architectures that replacing such network blocks with their corresponding stochastic solvers improves the accuracy of the model. We hope that the presented framework will contribute to a principled understanding of the theory and practice of deep network architectures.

**Acknowledgments.** This work was partially supported by the King Abdullah University of Science and Technology (KAUST) Office of Sponsored Research.

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

## A  LEAKY RELU AS A PROXIMAL OPERATOR

**Proposition 3.** The Leaky ReLU is the solution of the proximal operator of the function $g(\mathbf{x}) = \frac{\gamma}{2} \sum_i \max^2 (-\mathbf{x}_i - \lambda, 0)$ with slope $\frac{1}{1+\gamma}$ and shift $\lambda$.

*Proof.* The proximal operator is defined as

$$\text{Prox}_g(\mathbf{a}) = \arg\min_{\mathbf{x}} \frac{1}{2}\|\mathbf{x} - \mathbf{a}\|_2^2 + \underbrace{\frac{\gamma}{2} \sum_i \max^2 (-\mathbf{x}_i - \lambda, 0)}_{\gamma-\text{smooth}}$$

Note that the problem is both convex and smooth. The optimality conditions are given by:

$$\mathbf{x} - \mathbf{a} - \gamma\max(-\mathbf{x} - \lambda\mathbf{1}, 0) = 0$$

Since the problem is separable in coordinates, we have:

$$\mathbf{x}_i = \begin{cases} \mathbf{a}_i & \text{if } \mathbf{x}_i \geq -\lambda \\ \frac{\mathbf{a}_i - \gamma\lambda}{1+\gamma} & \text{if } \mathbf{x}_i \leq -\lambda \end{cases} \rightarrow \text{Prox}_g(\mathbf{a}_i) = \begin{cases} \mathbf{a}_i & \text{if } \mathbf{a}_i \geq -\lambda \\ \frac{\mathbf{a}_i - \gamma\lambda}{1+\gamma} & \text{if } \mathbf{a}_i \leq -\lambda. \end{cases}$$

The Leaky ReLU is defined as

$$\text{LeakyReLU}(\mathbf{a}_i) = \begin{cases} \mathbf{a}_i & \text{if } \mathbf{a}_i \geq 0 \\ \alpha\mathbf{a}_i & \text{if } \mathbf{a}_i \leq 0, \end{cases}$$

which shows that $\text{Prox}_g$ is a generalized form of the Leaky ReLU with a shift of $\lambda$ and a slope $\alpha = \frac{1}{1+\gamma}$. With $\lambda = 0$ and $\gamma = \frac{1-\alpha}{\alpha}$ the standard form of the Leaky ReLU is recovered. □

## B  APPROXIMATION OF A SOFT PLUS AS A PROXIMAL OPERATOR

**Proposition 4.** The proximal operator to $g(\mathbf{x}) = -\gamma \sum_i \log(\mathbf{x}_i)$ approximates the SoftPlus activation.

*Proof.* The proximal operator is defined as

$$\text{Prox}_g(\mathbf{a}) = \arg\min_{\mathbf{x}} \frac{1}{2}\|\mathbf{x} - \mathbf{a}\|_2^2 - \gamma \sum_i \log(\mathbf{x}_i). \tag{9}$$

Note that the function $g(\mathbf{x})$ is elementwise separable, convex, and smooth. By equating the gradient to zero and taking the positive solution of the resulting quadratic polynomial, we arrive at the closed-form solution:

$$\text{Prox}_g(\mathbf{a}) = \frac{1}{2}\mathbf{x} + \sqrt{\frac{1}{4}\mathbf{x} \odot \mathbf{x} + \gamma}, \tag{10}$$

where $\odot$ denotes elementwise multiplication. It is easy to see that this operator is close to zero for $\mathbf{x}_i \ll 0$, and close to $\mathbf{x}_i$ for $\mathbf{x}_i \gg 0$, with a smooth transition for small $|\mathbf{x}_i|$. □

Note that the function $\text{Prox}_g(\mathbf{a})$ approximates the activation $\text{SoftPlus} = \log(1 + \exp(\mathbf{a}))$ very well. An illustrative example is shown in Figure 2.

## C  APPROXIMATION OF TANH AND SIGMOID AS A PROXIMAL OPERATOR

**Proposition 5.** The proximal operator to $g(\mathbf{x}) = -\gamma \sum_i \log(1 - \mathbf{x}_i \odot \mathbf{x}_i)$ approximates the Tanh non-linearity.

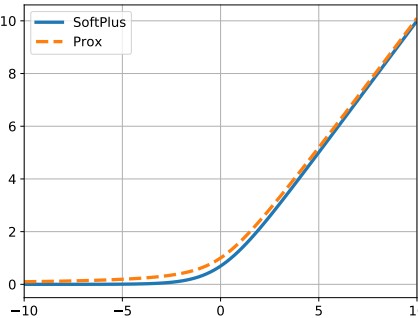 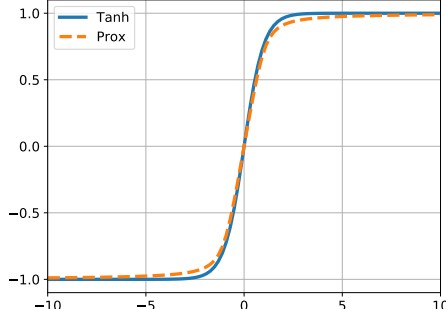

Figure 2: Comparison of common activation functions and their approximation via the corresponding proximal operator. Left: SoftPlus activation and corresponding proximal operator with $\gamma = 1$. Right: Tanh activation and corresponding proximal operator with $\gamma = 0.1$.

*Proof.* To simplify the exposition, we derive the proximal operator for the case $\gamma = 1$. The general case for $\gamma > 0$ follows analogously. The proximal operator is defined as

$$\text{Prox}_g(\mathbf{a}) = \arg\min_{\mathbf{x}} \frac{1}{2}\|\mathbf{x} - \mathbf{a}\|_2^2 - \sum_i \log(\mathbf{1} - \mathbf{x}_i \odot \mathbf{x}_i). \tag{11}$$

Note that the logarithm is taken element wise, and the objective is convex and smooth. By equating the gradient to zero, it can be seen that the optimal solution is a root of a cubic equation:

$$\mathbf{x}_i^3 - \mathbf{a}_i\mathbf{x}_i^2 - 3\mathbf{x}_i + \mathbf{a}_i = 0, \tag{12}$$

which is defined in each coordinate $i$ separately.

Let

$$p = \frac{\mathbf{a}_i^2}{9} + 1, \qquad q = \frac{\mathbf{a}_i^3}{27}, \tag{13}$$

Since $q^2 - p^3 < 0, \forall \mathbf{a}_i \in \mathbb{R}$, it is guaranteed that all roots are real and distinct. Consequently, the roots can be described as

$$r_k = 2\sqrt{p}\cos\left(\frac{1}{3}\cos^{-1}\left(\frac{\mathbf{a}_i^3}{27\sqrt{p^3}}\right) + \frac{2\pi k}{3}\right) + \frac{\mathbf{a}_i}{3}, \qquad k \in \{0, 1, 2\}. \tag{14}$$

Since $g(\mathbf{x})$ is only defined on $\mathbf{x} \in [-1, 1]^d$, the root that minimizes (11) has to satisfy

$$-1 \leq r_{opt} \leq 1. \tag{15}$$

$y = \cos^{-1}\left(\frac{\mathbf{a}_i^3}{27\sqrt{p^3}}\right)$ implies $0 \leq y \leq \pi$. Let

$$f_k = \cos\left(\frac{1}{3}y + \frac{2\pi k}{3}\right). \tag{16}$$

It is straightforward to check that

$$f_0 \in \left[\frac{1}{2}, 1\right], \qquad f_1 \in \left[-1, -\frac{1}{2}\right], \qquad f_2 \in \left[-\frac{1}{2}, \frac{1}{2}\right]. \tag{17}$$

By substituting $f_k$ into (14) and checking inequality (15) it becomes clear that the root corresponding to $k = 2$ minimizes (11). By using trigonometric identities the root corresponding to $k = 2$ can be further simplified to

$$(\text{Prox}_g(\mathbf{a}))_i = -2\sqrt{p}\sin\left(\frac{1}{3}\sin^{-1}\left(\frac{\mathbf{a}_i^3}{27\sqrt{p^3}}\right)\right) + \frac{\mathbf{a}_i}{3}, \tag{18}$$

which has the approximate shape of the Tanh activation. $\qquad\square$

An example of this operator is shown in Figure 2. The proximal operator corresponding to the Sigmoid activation can be derived in a similar fashion by setting $g(\mathbf{x}) = -\gamma \log(\mathbf{x}) - \gamma \log(\mathbf{1} - \mathbf{x})$.

## D    TENSOR PRELIMINARIES

The exposition presented in the paper requires some definitions related to tensors and operators on tensors. We summarize the material here. In all subsequent definitions, we assume $\mathcal{D} \in \mathbb{R}^{n_1 \times n_2 \times n_3 \times n_4}$ and $\vec{\mathcal{X}} \in \mathbb{R}^{n_2 \times 1 \times n_3 \times n_4}$.

**Definition D.1.** (Bibi & Ghanem, 2017) The t-product between high-order tensors is defined as

$$\mathcal{D} \circledast_{\text{HO}} \vec{\mathcal{X}} = \text{fold}_{\text{HO}} \left( \text{circ}_{\text{HO}} \left( \mathcal{D} \right) \text{MatVec}_{\text{HO}} \left( \vec{\mathcal{X}} \right) \right) \tag{19}$$

where $\text{circ}_{\text{HO}} \left( \mathcal{D} \right) \in \mathbb{R}^{n_1 n_3 n_4 \times n_2 n_3 n_4}$ and $\text{MatVec}_{\text{HO}} \left( \vec{\mathcal{X}} \right) \in \mathbb{R}^{n_2 n_3 n_4 \times 1}$.

The operator $\text{circ}_{\text{HO}}(.)$ unfolds an input tensor into a structured matrix. On the other hand, $\text{MatVec}_{\text{HO}}(.)$ unfolds an input tensor into a vector. The fold and unfold procedures are detailed in Bibi & Ghanem (2017).

**Definition D.2.** (Bibi & Ghanem, 2017) The operator

$$\text{bdiag} \left( \mathcal{D} \right) = \begin{bmatrix} \mathcal{D} \left( :, :, 1, 1 \right) & \mathbf{0}_{n_1 \times n_2} & \cdots & \cdots & \mathbf{0}_{n_1 \times n_2} \\ \cdots & \ddots & \ddots & \cdots & \vdots \\ \cdots & \cdots & \mathcal{D} \left( :, :, i, j \right) & \cdots & \vdots \\ \cdots & \ddots & \ddots & \ddots & \vdots \\ \cdots & \cdots & \cdots & \cdots & \mathcal{D} \left( :, :, n_3, n_4 \right) \end{bmatrix}, \tag{20}$$

where $\text{bdiag} \left( . \right) : \mathbb{C}^{n_1 \times n_2 \times n_3 \times n_4} \rightarrow \mathbb{C}^{n_1 n_3 n_4 \times n_2 n_3 n_4}$, maps a tensor to a block diagonal matrix of all the frontal faces $\mathcal{D}(:, :, i, j)$. Note that if $n_3 = n_4 = 1$, $\text{bdiag}()$ is an identity mapping. Moreover if $n_1 = n_2 = 1$, $\text{bdiag}(.)$ is a diagonal matrix.

Due to the structure of the tensor unfold of $\text{circ}_{\text{HO}}(.)$, the resultant matrix $\text{circ}_{\text{HO}}(\mathcal{D})$ exhibits the following blockwise diagonalization:

$$\text{circ}_{\text{HO}}(\mathcal{D}) = (\mathbf{F}_{n_4} \otimes \mathbf{F}_{n_3} \otimes \mathbf{I}_{n_1}) \, \text{bdiag} \left( \hat{\mathcal{D}} \right) (\mathbf{F}_{n_4} \otimes \mathbf{F}_{n_3} \otimes \mathbf{I}_{n_2})^{\mathbf{H}}, \tag{21}$$

where $\mathbf{F}_n$ is the $n \times n$ Normalized Discrete Fourier Matrix. Note that $\hat{\mathcal{D}}$ has the dimensions $n_3$ and $n_4$ replaced with the corresponding 2D Discrete Fourier Transforms. That is $\hat{\mathcal{D}}(i, j, :, :)$ is the 2D Discrete Fourier Transform of $\mathcal{D}(i, j, :, :)$.

For more details, the reader is advised to start with third order tensors in the work of Kilmer & Martin (2011) and move to the work of Bibi & Ghanem (2017) for extension to higher orders.

# E  PROOF OF LEMMA 1

Since non-linear activations commonly used in deep learning are elementwise, we only need to show that $\mathcal{A} \circledast_{\text{HO}} \vec{\mathcal{X}}^{l-1}$ performs a convolution. In particular we need to show, that $\mathcal{A} \circledast_{\text{HO}} \vec{\mathcal{X}}^{l-1}$ is equivalent to (**i**) performing 2D convolutions spatially along the third and fourth dimensions. (**ii**) It aggregates the result along the feature dimension $n_1$. (**iii**) It repeats the procedure for each of the $n_2$ filters independently. We will show the following using direct manipulation of the properties of $\circledast_{\text{HO}}$ (Bibi & Ghanem, 2017).

$$
\begin{aligned}
\mathcal{A} \circledast_{\text{HO}} \vec{\mathcal{X}}^{l-1} &= \sum_{i}^{n_1} \mathcal{A}(:,i,:,:) \circledast_{\text{HO}} \vec{\mathcal{X}}^{l-1}(i,:,:,:) \\
&= \sum_{i}^{n_1} \text{fold}_{\text{HO}} \left( \text{circ}_{\text{HO}}\left( (\mathcal{A}(:,i,:,:)) \right) \text{MatVec}\left( \vec{\mathcal{X}}^{l-1}(i,:,:,:) \right) \right)
\end{aligned}
\tag{22}
$$

Note that Equation (22) shows that features are aggregated along the $n_1$ dimension. Now, by showing that $\mathcal{A}(:,i,:,:) \circledast_{\text{HO}} \vec{\mathcal{X}}^{l-1}(i,:,:,:)$ performs $n_2$ independent 2D convolutions along on the $i^{\text{th}}$ channel, the Lemma 1 is proven. For ease of notation, consider two tensors $\vec{\mathcal{U}} \in \mathbb{R}^{n_2 \times 1 \times W \times H}$ and $\vec{\mathcal{Y}} \in \mathbb{R}^{1 \times 1 \times W \times H}$ then we have the following:

$$
\begin{aligned}
\vec{\mathcal{U}} \circledast_{\text{HO}} \vec{\mathcal{Y}} &\overset{(19)}{=} \text{fold}_{\text{HO}} \left( \text{circ}_{\text{HO}}\left( \vec{\mathcal{U}} \right) \text{MatVec}_{\text{HO}}\left( \vec{\mathcal{Y}} \right) \right) \\
&\overset{(21)}{=} \text{fold}_{\text{HO}} \left( (\mathbf{F}_H \otimes \mathbf{F}_W \otimes \mathbf{I}_{n_2}) \, \text{bdiag}\left( \vec{\hat{\mathcal{U}}} \right) \underbrace{(\mathbf{F}_H \otimes \mathbf{F}_W)^{\mathbf{H}}}_{\text{2D-Fourier Transform}} \text{MatVec}_{\text{HO}}\left( \vec{\mathcal{Y}} \right) \right) \\
&= \text{fold}_{\text{HO}} \left( (\mathbf{F}_H \otimes \mathbf{F}_W \otimes \mathbf{I}_{n_2}) \, \text{bdiag}\left( \vec{\hat{\mathcal{U}}} \right) \text{MatVec}_{\text{HO}}\left( \vec{\hat{\mathcal{Y}}} \right) \right)
\end{aligned}
$$

$$
\overset{(20)}{=} \text{fold}_{\text{HO}} \left( \underbrace{(\mathbf{F}_H \otimes \mathbf{F}_W \otimes \mathbf{I}_{n_2})}_{\text{2D-Inverse Fourier Transform with stride of of } n_2} \underbrace{\begin{bmatrix} \left[ \vec{\hat{\mathcal{U}}}(:,1,1,1) \right] \vec{\hat{\mathcal{Y}}}(1,1,1,1) \\ \vdots \\ \left[ \vec{\hat{\mathcal{U}}}(:,1,i,j) \right] \vec{\hat{\mathcal{Y}}}(1,1,i,j) \\ \vdots \\ \left[ \vec{\hat{\mathcal{U}}}(:,1,W,H) \right] \vec{\hat{\mathcal{Y}}}(1,1,W,H) \end{bmatrix}}_{\vec{\hat{\mathcal{G}}}} \right)
$$

Note that $\vec{\hat{\mathcal{G}}}$ is the elementwise product of the 2D Discrete Fourier Transform between a feature of an input activation $\vec{\mathcal{Y}}$ and the 2D Discrete Fourier Transform of every filter of the $n_2$ in $\vec{\mathcal{Y}}$. Since $(\mathbf{F}_H \otimes \mathbf{F}_W \otimes \mathbf{I}_{n_2})$ is the inverse 2D Fourier transform along each of the $n_2$ filters resulting in $\vec{\hat{\mathcal{G}}}$. Thus $\vec{\mathcal{U}} \circledast_{\text{HO}} \vec{\mathcal{Y}}$ performs 2D convolutions independently along each of the $n_2$ filters, combined with (22); thus, Lemma 1 is proven.

# F   PROOF OF LEMMA 2

Since,

$$L = \lambda_{\max}\left(\nabla^2 F(\vec{\mathcal{X}})\right) = \lambda_{\max}\left(\nabla^2\left(\frac{1}{n}\sum_i^{n_1}\|\mathcal{A}^{\mathbf{H}}(i,:,:,:) \circledast_{\text{HO}} \vec{\mathcal{X}} - \vec{\mathcal{X}}^{l-1}\|_F^2\right)\right)$$

$$\overset{(19)}{=} \lambda_{\max}\left(\nabla^2\left(\frac{1}{2}\sum_i^{n_1}\|\text{circ}_{\text{HO}}\left(\mathcal{A}^{\mathbf{H}}(i,:,:,:)\right)\text{MatVec}\left(\vec{\mathcal{X}}\right) - \text{MatVec}\left(\vec{\mathcal{X}}^{l-1}\right)\|_F^2\right)\right)$$

$$= \lambda_{\max}\left(\sum_i^{n_1}\text{circ}_{\text{HO}}^{\mathbf{H}}\left(\mathcal{A}^{\mathbf{H}}(i,:,:,:)\right)\text{circ}_{\text{HO}}\left(\mathcal{A}^{\mathbf{H}}(i,:,:,:)\right)\right)$$

$$= \lambda_{\max}\left(\text{circ}_{\text{HO}}^{\mathbf{H}}\left(\mathcal{A}^{\mathbf{H}}\right)\text{circ}_{\text{HO}}\left(\mathcal{A}^{\mathbf{H}}\right)\right). \tag{23}$$

The second equality also follows from the separability of $\|.\|_F^2$ where we dropped $\text{fold}_{\text{HO}}$. The last equality follows from the linearity of $\circledast_{\text{HO}}$. Moreover, from (19), we have:

$$\text{circ}_{\text{HO}}^{\mathbf{H}}\left(\mathcal{A}^{\mathbf{H}}\right)\text{circ}_{\text{HO}}\left(\mathcal{A}^{\mathbf{H}}\right)$$

$$\overset{(21)}{=} \left[\left(\mathbf{F}_H \otimes \mathbf{F}_W \otimes \mathbf{I}_{n_1}\right)\text{bdiag}\left(\hat{\mathcal{A}}^{\mathbf{H}}\right)\left(\mathbf{F}_H \otimes \mathbf{F}_W \otimes \mathbf{I}_{n_2}\right)^{\mathbf{H}}\right]^{\mathbf{H}}$$

$$\left[\left(\mathbf{F}_H \otimes \mathbf{F}_W \otimes \mathbf{I}_{n_1}\right)\text{bdiag}\left(\hat{\mathcal{A}}^{\mathbf{H}}\right)\left(\mathbf{F}_H \otimes \mathbf{F}_W \otimes \mathbf{I}_{n_2}\right)^{\mathbf{H}}\right]$$

$$= \left(\mathbf{F}_H \otimes \mathbf{F}_W \otimes \mathbf{I}_{n_2}\right)\text{bdiag}\left(\hat{\mathcal{A}}\right)\text{bdiag}\left(\hat{\mathcal{A}}^{\mathbf{H}}\right)\left(\mathbf{F}_H \otimes \mathbf{F}_W \otimes \mathbf{I}_{n_2}\right)^{\mathbf{H}}$$

$$\overset{(20)}{=} \left(\mathbf{F}_H \otimes \mathbf{F}_W \otimes \mathbf{I}_{n_2}\right)\begin{bmatrix}\hat{\mathcal{A}}(:,:,1,1)\hat{\mathcal{A}}^{\mathbf{H}}(:,:,1,1) & \mathbf{0} & \mathbf{0} \\ \mathbf{0} & \hat{\mathcal{A}}(:,:,1,2)\hat{\mathcal{A}}^{\mathbf{H}}(:,:,1,2) & \mathbf{0} \\ \vdots & \ddots & \vdots\end{bmatrix}\left(\mathbf{F}_H \otimes \mathbf{F}_W \otimes \mathbf{I}_{n_2}\right)^{\mathbf{H}}$$

$$= \left(\mathbf{F}_H \otimes \mathbf{F}_W \otimes \mathbf{I}_{n_2}\right)\underbrace{\begin{bmatrix}\mathbf{U}_{11} & \cdots & \mathbf{0} \\ \mathbf{0} & \mathbf{U}_{12} & \mathbf{0} \\ \mathbf{0} & \ddots & \cdots\end{bmatrix}}_{\text{eigenvectors}}\begin{bmatrix}\Sigma_{11} & \mathbf{0} & \mathbf{0} \\ \mathbf{0} & \Sigma_{12} & \mathbf{0} \\ & \ddots & \end{bmatrix}\begin{bmatrix}\mathbf{U}_{11}^{\mathbf{H}} & \cdots & \mathbf{0} \\ \mathbf{0} & \mathbf{U}_{12}^{\mathbf{H}} & \mathbf{0} \\ \mathbf{0} & \ddots & \cdots\end{bmatrix}\left(\mathbf{F}_H \otimes \mathbf{F}_W \otimes \mathbf{I}_{n_2}\right)^{\mathbf{H}}.$$

$$\tag{24}$$

The second equality follows from the orthogonality of $\mathbf{F}_H \otimes \mathbf{F}_W \otimes \mathbf{I}_{n_1}$. The fourth equality follows from $\hat{\mathcal{A}}(:,:,i,j)\hat{\mathcal{A}}^{\mathbf{H}}(:,:,i,j) = \mathbf{U}_{ij}\Sigma_{ij}\mathbf{U}_{ij}^{\mathbf{H}}$. Thus by combining (23) and (24), we have:

$$L = \lambda_{\max}\left(\nabla^2 F(\vec{\mathcal{X}})\right) = \lambda_{\max}\left(\begin{bmatrix}\Sigma_{11} & \mathbf{0} & \mathbf{0} \\ \mathbf{0} & \Sigma_{12} & \mathbf{0} \\ & & \ddots\end{bmatrix}\right) = \max_{i\in\{1,2,...,W\},j\in\{1,2,...,H\}}\left(\lambda_{\max}\left(\Sigma_{ij}\right)\right)$$

$$= \max_{i\in\{1,2,...,W\},j\in\{1,2,...,H\}}\left\{\lambda_{\max}\left(\hat{\mathcal{A}}(:,:,i,j)\hat{\mathcal{A}}^{\mathbf{H}}(:,:,i,j)\right)\right\}.$$

## G  PROOF OF PROPOSITION 1

**Lemma 3.** For $\tau$-nice Prox-SG,

$$\Psi_\tau(\vec{\mathcal{X}}) = \frac{1}{\tau}\left[\sum_{i \in S} \frac{n_1}{2} \|\mathcal{A}^{\mathbf{H}}(i,:,:,:) \circledast_{\text{HO}} \vec{\mathcal{X}} - \vec{\mathcal{X}}^{l-1}(i,:,:,:)\|_F^2 - \langle \vec{\mathcal{B}}, \mathcal{X} \rangle\right]$$

is an unbiased estimator to $F(\vec{\mathcal{X}})$.

*Proof.*

$$\mathbb{E}\left[\Psi_\tau(\vec{\mathcal{X}})\right] = \mathbb{E}\left[\frac{n_1}{\tau}\sum_{i \in S}\frac{1}{2}\|\mathcal{A}^{\mathbf{H}}(i,:,:,:) \circledast_{\text{HO}} \vec{\mathcal{X}} - \vec{\mathcal{X}}^{l-1}\|_F^2\right] - \frac{1}{\tau}\mathbb{E}\left[\sum_{i \in S}\langle \vec{\mathcal{B}}, \mathcal{X}\rangle\right]$$

$$= \frac{n_1}{2\tau}\sum_{i=1}^{n_1}\|\mathcal{A}^{\mathbf{H}}(i,:,:,:) \circledast_{\text{HO}} \vec{\mathcal{X}} - \vec{\mathcal{X}}^{l-1}\|_F^2 \mathbb{E}\left[\mathbb{1}_{i \in S}\right] - \frac{1}{\tau}\langle \vec{\mathcal{B}}, \mathcal{X}\rangle\sum_{i}^{n_1}\mathbb{E}\left[\mathbb{1}_{i \in S}\right]$$

$$= \frac{1}{2}\sum_{i}^{n_1}\|\mathcal{A}^{\mathbf{H}}(i,:,:,:) \circledast_{\text{HO}} \vec{\mathcal{X}} - \vec{\mathcal{X}}^{l-1}\|_F^2 - \langle \vec{\mathcal{B}}, \mathcal{X}\rangle = F(\vec{\mathcal{X}}).$$

The first equality follows by introducing an indicator function where $\mathbb{1}_{i \in S} = 1$ if $i \in S$ and zero otherwise. The last equality follows from the uniformity across elements of the $\tau$-nice $S$. $\qquad\square$

From Lemma 3, and with zero initialization it follows that

$$\nabla\Psi_\tau(\vec{\mathcal{X}})\Big|_{\vec{\mathcal{X}}=\mathbf{0}} = \nabla\left(\frac{1}{\tau}\sum_{i \in S}\frac{n_1}{2}\|\mathcal{A}^{\mathbf{H}}(i,:,:,:) \circledast_{\text{HO}} \vec{\mathcal{X}} - \vec{\mathcal{X}}^{l-1}(i,:,:,:)\|_F^2 - \langle \vec{\mathcal{B}}, \mathcal{X}\rangle\right)\Big|_{\vec{\mathcal{X}}=\mathbf{0}}$$

$$\stackrel{(19)}{=} \sum_{i \in S}\nabla\left(\frac{n_1}{2\tau}\|\text{circ}_{\text{HO}}\left(\mathcal{A}^{\mathbf{H}}(i,:,:,:)\right)\text{MatVec}\left(\vec{\mathcal{X}}\right) - \text{MatVec}\left(\vec{\mathcal{X}}^{l-1}(i,:,:,:)\right)\|_F^2 - \frac{1}{\tau}\langle \vec{\mathcal{B}}, \mathcal{X}\rangle\right)\Big|_{\vec{\mathcal{X}}=\mathbf{0}}$$

$$= \sum_{i \in S}\left(-\frac{n_1}{\tau}\text{circ}_{\text{HO}}^{\mathbf{H}}\left(\mathcal{A}^{\mathbf{H}}(i,:,:,:)\right)\text{MatVec}\left(\vec{\mathcal{X}}^{l-1}(i,:,:,:)\right) - \frac{1}{\tau}\vec{\mathcal{B}}\right)$$

$$= \sum_{i \in S}\left(-\frac{n_1}{\tau}\mathcal{A}(:,i,:,:) \circledast_{HO} \vec{\mathcal{X}}^{l-1}(i,:,:,:) - \frac{1}{\tau}\vec{\mathcal{B}}\right) \tag{25}$$

is an unbiased estimator of $\nabla F(\vec{\mathcal{X}})\big|_{\vec{\mathcal{X}}=\mathbf{0}}$. The last iteration follows by noting that $\mathcal{A} = \left(\mathcal{A}^{\mathbf{H}}\right)^{\mathbf{H}}$. Therefore, the first iteration of $\tau$-nice Prox-SGD with zero initialization and unit step size is:

$$\vec{\mathcal{X}} \leftarrow \text{Prox}_{\frac{1}{L}g}\left(-\nabla\Psi_\tau(\vec{\mathcal{X}})\big|_{\vec{\mathcal{X}}=\mathbf{0}}\right)$$

$$= \text{Prox}_{\frac{1}{L}g}\left(\sum_{i \in S}\left(\frac{n_1}{\tau}\mathcal{A}(:,i,:,:) \circledast_{HO} \vec{\mathcal{X}}^{l-1}(i,:,:,:) + \frac{1}{\tau}\vec{\mathcal{B}}\right)\right). \tag{26}$$

Note that the previous stochastic sum in (26) with $\tau = (1-p)n_1$ can be reparameterized as follows:

$$\vec{\mathcal{X}} = \text{Prox}_{\frac{1}{L}g}\left(\frac{1}{\tau}\vec{\mathcal{B}}\sum_{i}^{n_1}\mathbb{1}_{i \in S} + \frac{n_1}{\tau}\sum_{i}^{n_1}\mathcal{A}(:,i,:,:) \circledast_{HO} \vec{\mathcal{X}}^{l-1}(i,:,:,:)\mathbb{1}_{i \in S}\right)$$

$$= \text{Prox}_{\frac{1}{L}g}\left(\vec{\mathcal{B}} + \frac{n_1}{\tau}\mathcal{A}\circledast_{HO}\left(\mathcal{M} \odot \vec{\mathcal{X}}^{l-1}\right)\right) = \text{Prox}_{\frac{1}{L}g}\left(\vec{\mathcal{B}} + \frac{n_1}{\tau}\mathcal{A}\circledast_{HO}\text{BerDropout}_p\left(\vec{\mathcal{X}}^{l-1}\right)\right) \tag{27}$$

where $\mathcal{M} \in \mathbb{R}^{n_1 \times 1 \times W \times H}$ is a mask tensor. Note that since $\tau = (1-p)n_1$, $\mathcal{M}$ has exactly $pn_1$ slices $\mathcal{M}(i, :, :, :)$ that are completely zero. This equivalent to a dropout layer where the layer drops exactly $pn_1$ input activations. It follows that (27) is equivalent to a forward pass through a BerDropout$_p$ layer followed by a linear layer and non-linear activation.

## H    PROOF OF PROPOSITION 2

mS2GD (Konečnỳ et al., 2016) with zero initialization at the first epoch defines the following update:

$$\vec{\mathcal{X}} \leftarrow \text{Prox}_{\frac{1}{L}g} \left( \vec{\mathcal{X}} - \nabla F(\vec{\mathcal{Y}}) - \left( \frac{1}{\tau} \sum_{i \in S} \left( \nabla f_i(\vec{\mathcal{X}}) - \nabla f_i(\vec{\mathcal{Y}}) \right) \right) \right). \tag{28}$$

With zero initialization at the first epoch we have $\mathcal{Y} = \mathbf{0}$, therefore

$$\nabla F(\vec{\mathcal{Y}})|_{\vec{\mathcal{Y}}=\mathbf{0}} = \frac{1}{n_1} \sum_{i}^{n_1} \nabla f_i(\mathcal{Y})|_{\vec{\mathcal{Y}}=\mathbf{0}} \overset{(25)}{\underset{\tau=n_1}{=}} -\mathcal{A} \circledast_{\text{HO}} \vec{\mathcal{X}}^{l-1} - \mathcal{B}. \tag{29}$$

Moreover,

$$\frac{1}{\tau} \sum_{i \in S} \nabla f_i(\vec{\mathcal{X}}) - \frac{1}{\tau} \sum_{i \in S} \nabla f_i(\mathcal{Y})|_{\vec{\mathcal{Y}}=\mathbf{0}}$$

$$= \frac{1}{\tau} \sum_{i \in S} n_1 \text{circ}_{\text{HO}}^{\mathbf{H}} \left( \mathcal{A}^{\mathbf{H}}(i, :, :, :) \right) \left[ \text{circ}_{\text{HO}} \left( \mathcal{A}^{\mathbf{H}}(i, :, :, :) \right) \text{MatVec} \left( \vec{\mathcal{X}} \right) - \text{MatVec} \left( \vec{\mathcal{X}}^{l-1} \right) \right] - \vec{\mathcal{B}}$$

$$- \left[ \frac{1}{\tau} \sum_{i \in S} -n_1 \text{circ}_{\text{HO}} \left( \mathcal{A}^{\mathbf{H}}(i, :, :, :) \right)^{\mathbf{H}} \text{MatVec} \left( \vec{\mathcal{X}}^{l-1} \right) - \vec{\mathcal{B}} \right]$$

$$= \frac{n_1}{\tau} \sum_{i \in S} \text{circ}_{\text{HO}}^{\mathbf{H}} \left( \mathcal{A}^{\mathbf{H}}(i, :, :, :) \right) \text{circ}_{\text{HO}} \left( \mathcal{A}^{\mathbf{H}}(i, :, :, :) \right) \text{MatVec} \left( \vec{\mathcal{X}} \right)$$

$$= \frac{n_1}{\tau} \sum_{i \in S} \mathcal{A}(:, i, :, :) \circledast_{\text{HO}} \mathcal{A}^{\mathbf{H}}(i, :, :, :) \circledast_{\text{HO}} \vec{\mathcal{X}} \tag{30}$$

Substituting both (29) and (30) into (28) we have

$$\vec{\mathcal{X}} \leftarrow \text{Prox}_{\frac{1}{L}g} \left( \vec{\mathcal{X}} + \mathcal{A} \circledast_{\text{HO}} \vec{\mathcal{X}}^{l-1} + \vec{\mathcal{B}} - \left( \frac{n_1}{\tau} \sum_{i \in S} \mathcal{A}(:, i, :, :) \circledast_{\text{HO}} \mathcal{A}^{\mathbf{H}}(i, :, :, :) \circledast_{\text{HO}} \vec{\mathcal{X}} \right) \right)$$

$$= \text{Prox}_{\frac{1}{L}g} \left( \underbrace{\mathcal{A} \circledast_{\text{HO}} \vec{\mathcal{X}}^{l-1} + \vec{\mathcal{B}}}_{\text{Fully-connected/Convolutional}} + \left( \mathcal{I} - \frac{n_1}{\tau} \sum_{i \in S} \mathcal{A}(:, i, :, :) \circledast_{\text{HO}} \mathcal{A}^{\mathbf{H}}(i, :, :, :) \right) \circledast_{\text{HO}} \vec{\mathcal{X}} \right) \tag{31}$$

Note that $\mathcal{I} \in \mathbb{R}^{n_2 \times n_2 \times W \times H}$ is the identity tensor in which all frontal slices are identity $\mathcal{I}(:, :, i, j) = \mathbf{I}_{n_2 \times n_2}$ and all other slices are zeros. Following the definition in (19), we have $\mathcal{I} \circledast_{\text{HO}} \vec{\mathcal{X}} = \vec{\mathcal{X}}$.

## I  RANDOMIZED COORDINATE DESCENT PERMUTES DROPOUT AND LINEAR LAYER

We present an additional insight to the role of stochastic solvers on (5) in network design. In particular, we show that performing a randomized coordinate descent, RCD, on (5) ignoring the finite sum structure, is equivalent to a linear transformation followed by BerDropout$_p$ and a non-linear activation. That is, performing RCD permutes the order of linear transformation and dropout. For ease of notation, we show this under the special case of fully connected layers.

**Proposition 6.** A single iteration of Randomized Coordinate Descent, e.g. NSync (Richtárik & Takáč, 2016), with $\tau$-nice sampling of coordinates of (5) with $\tau = (1-p)n_2$, unit step sizes along each partial derivative, and with zero initialization is equivalent to:

$$\text{Prox}_{\frac{1}{L}g}\left(\sum_{i \in S} \mathcal{A}(i,:,:,:) \circledast_{\text{HO}} \vec{\mathcal{X}}^{l-1} + \mathcal{B}(i,:,:,:)\right),$$

which is equivalent to a forward pass through a linear layer followed by a BerDropout$_p$ layer (that drops exactly $n_2 p$ output activations) followed by a non-linear activation.

*Proof.* We provide a sketch of the proof on the simple quadratic $F(\mathbf{x}) = \frac{1}{2}\|\mathbf{A}^\top \mathbf{x} - \mathbf{x}^{l-1}\|^2 - \mathbf{b}^\top \mathbf{x}$ where the linear layer is a fully-connected layer. Considering a randomized coordinate descent, e.g. NSync, with $\tau$-nice sampling of the coordinates we have the following:

$$\mathbf{x} \leftarrow \text{Prox}_g\left(\mathbf{x} - \sum_{i \in S} \frac{1}{v_i} \mathbf{e}_i^\top \nabla F(\mathbf{x}) \mathbf{e}_i\right)$$

$$= \text{Prox}_g\left(\mathbf{x} - \sum_{i \in S} \frac{1}{v_i} \mathbf{e}_i^\top \left(\mathbf{A}\left(\mathbf{A}^\top \mathbf{x} - \mathbf{x}^{l-1}\right) - \mathbf{b}\right) \mathbf{e}_i\right)$$

$$\overset{\mathbf{x}=0}{=} \text{Prox}_g\left(-\sum_{i \in S} \frac{1}{v_i} \mathbf{e}_i^\top \left(-\mathbf{A}\mathbf{x}^{l-1} - \mathbf{b}\right) \mathbf{e}_i\right)$$

$$= \text{Prox}_g\left(\sum_{i \in S} \mathbf{e}_i^\top \left[\text{Diag}\left(\mathbf{v}\right)^{-1}\mathbf{A}\mathbf{x}^{l-1} + \mathbf{b}\right] \mathbf{e}_i\right)$$

$$\overset{\mathbf{v}=\mathbf{1}}{=} \text{Prox}_g\left(\text{Dropout}_p\left(\mathbf{A}\mathbf{x}^{l-1} + \mathbf{b}\right)\right) \tag{32}$$

Note that $\mathbf{e}_i$ is a vector of all zeros except the $i^{\text{th}}$ coordinate which is equal to 1. Moreover, since the step sizes along each partial derivative is 1, $\mathbf{v} = \mathbf{1}$. Equation (32) is equivalent to a forward pass through a linear layer followed by a BerDropout$_p$ layer and a non-linear activation. $\square$

## J   RESNET BLOCKS

In here, we briefly discuss how a ResNet block can be incorporated as an iteration of a solver. The key observation is that a ResNet block has two non-linearities connected through a skip connection. In particular, consider the $l^{th}$ ResNet block with two linear operators denotes as $\mathcal{A}^{(1)}$ and $\mathcal{A}^{(2)}$ with biases as $\mathcal{B}^{(1)}$ and $\mathcal{B}^{(2)}$, respectively. The input activations to the ResNet block form the previous layer. Therefore, a forward pass through a ResNet block are $\mathcal{X}^{k-1}$ where $\mathcal{X}^l$ are the output activations. A forward pass through a ResNet block exhibits the following functional form:

$$\mathcal{X}^l = \text{ReLU}\left(\mathcal{A}^{(2)} \circledast_{\text{HO}} \left(\text{ReLU}\left(\mathcal{A}^{(1)} \circledast_{\text{HO}} \mathcal{X}^{l-1} + \mathcal{B}^{(1)}\right)\right) + \mathcal{B}^{(2)} + \mathcal{X}^{(l-1)}\right)$$

This can be seen as two blocks of linear followed by nonlinear composed such that:

$$\tilde{\mathcal{X}} = \text{ReLU}\left(\mathcal{A}^{(1)} \circledast_{\text{HO}} \mathcal{X}^{l-1} + \mathcal{B}^{(1)}\right)$$
$$\mathcal{X}^l = \text{ReLU}\left(\mathcal{A}^{(2)} \circledast_{\text{HO}} \tilde{\mathcal{X}} + \mathcal{B}^{(2)} + \mathcal{X}^{l-1}\right)$$

Note that a a batch normalization layer is linear during the forward pass through the network and thus can be absorbed in $\mathcal{A}$ and $\mathcal{B}$. Consequently, a forward pass through a ResNet block can be viewed as performing a single iteration of two consecutive stochastic or deterministic solvers (depending on the presence or absence of Dropout layers) on (5).

