# OpenReview forum: "Deep Layers as Stochastic Solvers"
_ICLR.cc/2019/Conference_

### Official Review · AnonReviewer1 · 2018-11-01
**Interesting paper, should be accepted**

**Rating:** 8
**Confidence:** 1

**Review:**

This paper presents a very interesting interpretation of the neural network architecture.

I think what is remarkable is that the author presents the general results (beyond the dense layer) including a convolutional layer by using the higher-order tensor operation.
Also, this research gives us new insight into the network architecture, and have the potential which leads to many interesting future directions.
So I think this work has significant value for the community.

The paper is clearly written and easy to follow in the meaning that the statement is clear and enough validation is shown. (I found some part of the proof are hard to follow.)

\questions
In the experiment when you mention about "embed solvers as a replacement to their corresponding blocks of layers", I wonder how they are implemented. About the feedforward propagation, I guess that for example, the prox operator is applied multiple times to the input, but I cannot consider what happens about the backpropagation of the loss.

In the experiment, the author mentioned that  "what happens if the algorithm is applied for multiple iterations?". From this, I guess the author iterate the corresponding algorithms several times, but actually how many times were the iterations or are there any criterion to stop the algorithm?

\minor comments
The definition of \lambda_max below Eq(3) are not shown, thus should be added.

---

> ### Author Response · Authors · 2018-11-13
> **Response To R1**
>
> We thank R1 for the positive comments and review.
>
> (1) Definition of $\lambda_{\max}$ in Eq (3).
>
> We have adjusted the text below Eq (3) to clearly state that $\lambda_{\text{max}}(.)$ is the maximum eigenvalue function.
>
> (2) On the implementation of solvers.
>
> The block of layers (dropout followed by linear and nonlinear layers) is replaced with an iterative solver, i.e. recurrent layer, that performs the Prox operator several times before generating an output. This is for the forward pass through the network. As for backpropagation, it is performed by simply unrolling the layers. This is commonly referred to as backpropagation through time in recurrent neural networks. While this is not particularly efficient in general, there are several potential ways to improve this by taking gradients implicitly through the argmin operator [1,2]. We leave this to future work.
>
> (3) On the number of iterations.
>
> The number of iterations was always kept constant. It was  set to 10 as stated at the end of page 6 for all small networks. As for larger networks such as VGG16, this number is fixed to 30 iterations as discussed in page 7 (just below Table 3). At present we do not have a universal criterion for choosing the number of iterations; this is treated as a hyperparameter.
>
>
> [1] ``Techniques for Gradient-Based Bilevel Optimization with Non-smooth Lower Level Problems", Peter Ochs, René Ranftl, Thomas Brox, Thomas Pock.
>
> [2] ``On Differentiating Parameterized Argmin and Argmax Problems with Application to Bi-level Optimization", Stephen Gould, Basura Fernando, Anoop Cherian, Peter Anderson, Rodrigo Santa Cruz, Edison Guo.

---

### Official Review · AnonReviewer2 · 2018-11-04
**Review comments on “Deep Layers as Stochastic Solvers”**

**Rating:** 7
**Confidence:** 5

**Review:**

This paper theoretically verifies an equivalence between stochastic solvers on a particular class of convex optimization problems and a forward pass through a dropout layer followed by a linear layer and a non-linear activation. Experiments show that replacing a block of layers with multiple iterations of the corresponding solver improves classification accuracy. My detailed comments are as follows.

*Positive points:

1. The perspective is novel and interesting, i.e., training a forward pass through a dropout layer followed by a linear layer and a non-linear activation is equivalent to optimizing a convex problem by a Proximal Stochastic Gradient method. More importantly, this perspective has been theoretically verified.

2. In the experiments, training networks with solvers replacing deep layers is able to improve accuracy significantly.

*Negative points:

1. Some technical details are not clear and many notations are used without clear explanations. Specifically, many notations based on (Bibi & Ghanem, 2017) make the paper hard to follow. Moreover, there are many mistakes in proofs. Please revise the paper according to the following comments.

2. There are many limitations for the proposed method. Specifically, the theoretical results are hard to be extended to more general neural networks (e.g., ResNet) with Batch Normalization which are widely used.

3. The experiment section should be significantly improved. There are only two datasets (i.e., CIFAR-10, CIFAR-100). It would be convincing that more baselines are compared on other datasets, such as ImageNet.

*Detailed comments:

**Comments on technical issues.

1. In Problem (1), the definition of $g(x)$ and $f¬_i()$ should be provided for clarity.

2. The motivation and some details of Function (2) should be provided since $F(x^l)$ is important for proving the equivalence between stochastic solvers and a forward network. In addition, $x$ should be corrected as $x^l$.

3. Is Equation (3) wrong? Based on the definition of Prox-GD in (Xiao & Zhang, 2014), it should be $x^l=Prox(x^{l-1} – 1/L \nabla F(x^l)) = Prox((I-1/L A)x^{l-1} + 1/L (AA^T x^l + b))$ which is different from Equation (3). Moreover, the Lipschitz constant w.r.t. maximal eigenvalue should be proved.

4. In Definitions D.1 and D.2, what is the definition of $fold_{H0}$? Is the dimensionality of $bdiag(D)$ wrong? Why is $bdiag(D)$ an identity mapping when $n_3=n_4$?

5. There are some issues on Equation (7) and its proofs. Is $A(:, i, :, :)$ and $\vec{X}(i, :, :, :) $ wrong? It affects the results of Equation (8). Does Equation (25) miss the operator $fold_{HO}$ in Appendix G? Please check the proofs of Proposition 1.

6. There are some issues on proofs of Lemma 2. Why are $F_H \otimes F_W \otimes I_{n_1}$ and $F_H \otimes F_W \otimes I_{n_2}$ orthogonal? Is the third and fourth equality in (24) wrong? For the fourth equality in (24), Eigen decomposition seems to be for a matrix, not a tensor.


**Comments on Experiments

1. Training Networks is equivalent to optimizing proximal solvers. Why can training networks with solvers replacing blocks of layers improve accuracy? Reasonable explanations should be provided.

2. Optimizing a convex optimization problem can easily obtain the optimal solution. What happens if solvers are used to replace more blocks of layers? Complexity analysis for these should be provided.

3. The experiments are only conducted on two datasets (i.e., CIFAR-10, CIFAR-100). It would be better to compared more baselines on other datasets, such as ImageNet.

---

> ### Author Response · Authors · 2018-11-13
> **Response to R2 (1/4)**
>
> * We recommend reading this response in the revised PDF uploaded to OpenReview. This response is in Appendix L. The mathematical notation is easier to read in the PDF.
>
> We thank R2 for the comments and the positive feedback on the novelty of our approach.
>
> R2 raised several issues regarding the proofs. We have proofread all the proofs and there are no factual errors. In fact, some of the main results, e.g. Lemma 2, have also been verified numerically. However, there were some minor typos and non-standard notation that may have been confusing. We have corrected these typos in the revised version uploaded to OpenReview. Below is a detailed answer to all of R2's concerns.
>
>
> General Comments.
>
> (1) Unclear technical details and mistakes in the proofs.
>
> There are no mistakes in the proofs. We have thoroughly checked them and they are all correct. There were some typos that may have been behind R2's confusion. We have corrected these typos and improved the notation.
>
> (2) Limitations of the current method. What about ResNets and BatchNorm?
>
> Our framework can be directly applied to ResNets. A ResNet block can be viewed as two consecutive stochastic solvers. We have added Appendix J discussing this in the supplementary material. This approach is somewhat simplistic; there is room for exciting future work. Normalization layers are easy to handle. Note that during test time normalization layers are linear; thus they can be combined with the fully-connected or convolutional layer as a single linear layer.

---

> > ### Author Response · Authors · 2018-11-13
> > **Continuation to the Response to R2 (2/4)**
> >
> > Detailed Comments.
> >
> > (1) Definition of $g(\mathbf{x})$ and $f_i(\mathbf{x})$.
> >
> > The structure in problem (1) is standard in the machine learning and optimization communities [1,2,3,4]. As suggested by R2 and R3, we have added further details and examples in the first page. Note that for instance, if $f_i(\mathbf{a}_i^\top \mathbf{x}) = \frac{1}{2}(\mathbf{a}_i^\top \mathbf{x} - \mathbf{y}_i)^2$ and $g(\mathbf{x}) = \frac{1}{2}\|\mathbf{x}\|_2^2$, we recover ridge regression, while with $g(\mathbf{x}) = \|\mathbf{x}\|_1$ we recover LASSO regression.
> >
> > (2) The motivation and some details of Function (2). In addition, $x$ should be corrected as $x^l$.
> >
> > The missing superscript was corrected. It is not clear to us what R2 is asking for in terms of extra motivation and details. Could you clarify, so that we can alleviate this concern in the paper?
> >
> > (3) Is Equation (3) wrong?
> >
> > No, Eq (3) is correct. Note that $\mathbf{x}^l$ is the optimization variable while $\mathbf{x}^{l-1}$ is a fixed optimization parameter. R2 is confusing $\mathbf{x}^{l-1}$ (the activation output of the $l-1$ layer) with the optimization variable  $\mathbf{x}^l$ (the output activations of layer $l$). Thus, unlike what is stated by R2, the Prox-GD update has the following form: $\mathbf{x}^l \leftarrow \text{Prox} \left(\mathbf{x}^l - \frac{1}{L} \nabla F(\mathbf{x}^l)\right)$.  Consequently, we have $\mathbf{x}^l \leftarrow \text{Prox} \left(\mathbf{x}^l - \frac{1}{L}\left(\mathbf{A}\mathbf{A}^\top\mathbf{x}^l - \mathbf{A} \mathbf{x}^{l-1} - \mathbf{b}\right)\right)$, which is identical to Eq (3). Lastly, R2 is asking to prove ``the Lipschitz constant w.r.t. maximal eigenvalue''. We do not understand the request. It is well-known that the maximum eigenvalue of the Hessian of $F(\mathbf{x}^l)$ induces the tightest quadratic upper bound for the smooth part $F(\mathbf{x})$. This leads to the optimal largest step size $\frac{1}{L}$ in Prox-GD. This is well-known and straightforward to show [5].
> >
> > (4) Definition of $\text{fold}_{\text{H0}}(.)$? Is the dimensionality of $\text{bdiag}(\mathcal{D})$ wrong? Why is $\text{bdiag}(\mathcal{D})$ an identity mapping when $n_3=n_4$?
> >
> > a) The definition of $\text{fold}_{\text{HO}}(.)$ can be found in [11] and in [12] for third-order tensors. The operator $\text{fold}_{\text{H0}}(.)$ reshapes the elements of a matrix into a tensor. The precise reshape procedure can be best described as the inverse reshape to the tensor-to-matrix unfold operator $\text{MatVec}_{\text{HO}}(.)$ such that the following holds: $\text{fold}_{\text{HO}} \left(\text{MatVec}_{HO}\left(\mathcal{A}\right)\right)= \mathcal{A}$. For example, for a third-order tensor $\mathcal{A} \in \mathbb{R}^{n_1 \times n_2 \times n_3}$, $\text{MatVec}_{\text{HO}}\left(\mathcal{A}\right) \in \mathbb{R}^{n_1 n_3 \times n_2}$, which is a matrix, while $\text{fold}_{\text{HO}} \left(\text{MatVec}_{HO}\left(\mathcal{A}\right)\right)$ reshapes that matrix into the original tensor $\mathcal{A} \in \mathbb{R}^{n_1 \times n_2 \times n_3}$.
> >
> > b) Regarding $\text{bdiag}(\mathcal{D})$, there are indeed two typos. The dimensionality of $\text{bdiag}(\mathcal{D})$ is $\mathbb{C}^{n_1 n_3 n_4 \times n_2 n_3 n_4}$. If $n_3 = n_4 = 1$ then $\text{bdiag}(\mathcal{D})$ is an identity mapping. We have corrected the typos in the revision.
> >
> > (5) Some issues in Eqs (7,8). Does Eq (25) miss the operator $\text{fold}_{\text{H0}(.)}$ in Appendix G?
> >
> > Eqs (7,8,25) are correct with nothing missing. Eq (25) is not missing any operator. The reason $\text{fold}_{\text{HO}}(.)$ did not appear in Eq (25) simply follows from the fact that the Frobenius norm of a tensor and the Frobenius norm of the unfolded tensor are identical since $\text{fold}_{\text{HO}}(.)$ performs reordering of the elements. That is, $\|\text{MatVec}_{\text{HO}} \left(\mathcal{A}\right)\|_F^2 = \|\text{fold}_{\text{HO}}\left(\text{MatVec}_{\text{HO}} \left(\mathcal{A}\right)\right)\|_F^2$. This has also been discussed in the text below Eq (23) as this identity appears there too.

---

> > > ### Author Response · Authors · 2018-11-13
> > > **Continuation to the Response to R2 (3/4)**
> > >
> > > (6) On the proof of Lemma 2 and the orthogonality of $\mathbf{F}_H \otimes \mathbf{F}_W \otimes \mathbf{I}_{n_1}$. Is the third and fourth equalities in (24) wrong? On the fourth equality in (24).
> > >
> > > The proof in Lemma 2 is correct. First, note that the normalized DFT matrices $\mathbf{F}_{H}$ and $\mathbf{F}_W$ and the identity matrix $\mathbf{I}$ are orthogonal/unitary. It is trivial to prove using properties of the Kronecker product that the Kronecker product of orthogonal matrices is an orthogonal matrix. Note that for matrices $\mathbf{A}, \mathbf{B},\mathbf{C},\mathbf{D}$ of appropriate sizes, $(\mathbf{A} \otimes \mathbf{B})^\top = \mathbf{A}^\top \otimes \mathbf{B}^\top$ and $(\mathbf{A} \otimes \mathbf{B}) (\mathbf{C} \otimes \mathbf{D}) = \mathbf{A} \mathbf{C} \otimes \mathbf{B} \mathbf{D}$. Using the previous properties, the proof follows trivially. The only typo is in the text below Eq (24). $\mathcal{\hat{A}}(:,:,i,j) \mathcal{\hat{A}}^{\textbf{H}}(:,:,i,j) = \mathbf{U}_{ij} \Sigma_{ij} \mathbf{U}^{\mathbf{H}}_{ij}$. That is, we are performing an eigendecomposition of the faces of the tensors which are matrices. We have corrected the typo and changed the notation a bit in Eq (24) for further clarity.
> > >
> > > Comments on Experiments.
> > >
> > > (1) Training Networks is equivalent to optimizing proximal solvers. Why can training networks with solvers replacing blocks of layers improve accuracy? Reasonable explanations should be provided.
> > >
> > > In general, we do not have theoretical generalization bounds justifying the improvement. However, similar observations [6,7] have been made in various contexts upon designing networks. It seems that the recurrent structure in hidden activations has merits in improving generalization but no one has thorough theoretical foundations for this yet. Such behaviour has also been observed very recently in [8] where ODE solvers are used rather than optimization solvers. Our hypothesis is that since the output of a feed-forward network approximates the optimal solution of some convex optimization in each layer, attaining better solutions by performing more iterations of the corresponding solver may improve accuracy.
> > >
> > >
> > > (2) Optimizing a convex objective can easily obtain the optimal solution. What happens if solvers are used to replace more blocks of layers? Complexity analysis for these should be provided.
> > >
> > > Optimizing convex objectives is easier than other problems in the sense of global optimality guarantees and analysis of various deterministic and stochastic algorithms. However, achieving high accuracy solutions may still require performing a large number of iterations (e.g., $>10^3$). In our framework, larger number of iterations will increase the computational complexity of the network, particularly during training. On the other hand, replacing more blocks of layers with solvers (each with a small number of iterations) has the potential to improve the performance as highlighted in Table (4). However, replacing more blocks of feed-forward networks with solvers will suffer from diminishing returns, i.e. the improvements will tend to be smaller with more blocks replaced with solvers. This is firstly because many networks are already close to saturating datasets (e.g., VGG16 on CIFAR10). That is to say that the baseline networks are already achieving high accuracy ($> 90\%$). Secondly, some networks do not have enough capacity to improve on certain datasets (e.g., AlexNet on CIFAR-100). Note that replacing blocks of layers with solvers does not increase capacity; thus the improvement is attributed to the new structure and not to any form of over-parameterization of the network. As for complexity, adding $t$ layers of a solver is as expensive as performing $t$ feed-forward passes through a single layer.

---

> > > > ### Author Response · Authors · 2018-11-13
> > > > **Continuation to the Response to R2 (4/4)**
> > > >
> > > > (3) The experiments are only conducted on two datasets (i.e., CIFAR-10, CIFAR-100). It would be better to compare more baselines on other datasets, such as ImageNet.
> > > >
> > > > We were mostly interested in the theory around connecting stochastic solvers to feed-forward passes through networks and generalizing linear layers through tensor theory, which will potentially inspire the design of new dropout layers. Many machine learning and optimization papers conduct experiments on MNIST only, or MNIST and CIFAR-10 [8,9,10]. We have conducted experiments on both CIFAR-10 and CIFAR-100, which match or exceed the complexity of datasets used in multiple comparable papers published in NIPS/ICML/ICLR in recent years. We do not think that it is necessary for our work to conduct experiments on ImageNet.
> > > >
> > > >
> > > > References:
> > > >
> > > > [1] "Minimizing Finite Sums with the Stochastic Average Gradient", Mark Schmidt, Nicolas Le Roux, Francis Bach.
> > > > [2] "Stochastic Dual Coordinate Ascent Methods for Regularized Loss
> > > > Minimization" JMLR14, Shai Shalev-Shwartz, Tong Zhang.
> > > > [3] "SAGA: A Fast Incremental Gradient Method With Support for Non-Strongly Convex Composite Objectives", NIPS14, Aaron Defazio, Francis Bach, Simon Lacoste-Julien.
> > > > [4] "Stochastic Proximal Gradient Descent with Acceleration Techniques". NIPS14, Atsushi Nitanda.
> > > > [5] "Introductory Lectures on Convex Optimization: A Basic Course (Applied Optimization)". Kluwer Academic Publishers, 2004, Yuri Nesterov.
> > > > [6] "On Multi-Layer Basis Pursuit, Efficient Algorithms and Convolutional Neural Networks", arXiv 2018, Jeremias Sulam, Aviad Aberdam and Michael Elad.
> > > > [7] "ISTA-Net: Iterative Shrinkage-Thresholding Algorithm Inspired Deep Network for Image Compressive Sensing", CVPR18, Jian Zhang and Bernard Ghanem.
> > > > [8] "Neural Ordinary Differential Equations", NIPS18, Ricky T. Q. Chen, Yulia Rubanova, Jesse Bettencourt, David Duvenaud.
> > > > [9] "Variational Dropout and the Local Reparameterization Trick", NIPS15, Diederik P. Kingma, Tim Salimans, and Max Welling.
> > > > [10] "Variational Dropout Sparsifies Deep Neural Networks", ICML17, Dmitry Molchanov, Arsenii Ashukha, Dmitry Vetrov.
> > > > [11] "High order tensor formulation for convolutional sparse coding", ICCV17, Adel Bibi and Bernard Ghanem.
> > > > [12] "Factorization strategies for third-order tensors", Linear Algebra and its Applications 2011, Misha Kilmer, Carla Martin.

---

> > > > > ### Comment · AnonReviewer2 · 2018-11-29
> > > > > **response to author feedback**
> > > > >
> > > > > I highly appreciate the author response to my reviews. I think most of my concerns have been addressed.

---

### Official Review · AnonReviewer3 · 2018-11-08
**Review of Deep Layers as Stochastic Solvers**

**Rating:** 7
**Confidence:** 4

**Review:**

Overview:  This paper shows that the forward pass of a fully-connected layer (generalized to convolutions) followed by a nonlinearity in a neural network is equivalent to an iteration of a prox algorithm, where different regularizers in the objective of the related prox problem correspond to different nonlinearities such as ReLu. This connection is quite interesting. They further relate different stochastic prox algorithms to different dropout  layers and show results of improved performance on CIFAR-10 and CIFAR-100 on several architectures. The paper is well-written.

Major Concerns:

1. While the equivalence of one iteration of a prox algorithm and a single forward pass of the block is understandable, it is not clear what happens from making several iterations (10 in the case of fully-connected layers in the experiments) of the prox algorithm. It seems that this would be equivalent to making a forward pass through 10 equivalent blocks (i.e., 10 layers with the same weights and biases). But then the backward pass is still through the original network, so the problem being solved is not clear. Clarity on this would help.

2. Since the equivalence of 10 forward passes of a block are done at each iteration, using solvers does more computations (can be thought of as extra forward passes through extra layers as noted above), which makes the comparison not completely fair. Either adding more batches or more passes over the same batch multiple times (or at least for a few batches just to use the some computational power) would be more fair and likely improve the performance of the baseline networks.

Minor Issues:

1. missing definitions such as g(x) at beginning of Section 3 and p in Proposition 1.

2. Give examples of where the prox problems in Table 1 show up in practice (outside of activation functions in neural networks)

3. It says "for different choices of dropout rate the baseline can always be improved by..." in the Experiments.  This is not provable.

4. Include results for Dropout rate p=0 in Table 5.

---

> ### Author Response · Authors · 2018-11-13
> **Response to R3**
>
> * We recommend reading this response in the revised PDF uploaded to OpenReview. This response is in Appendix M. The mathematical notation is easier to read in the PDF.
>
> We thank R3 for the positive review and feedback. Below are our responses to all concerns.
>
> On the major concerns.
>
> (1) Some clarity on the forward/backward pass of networks with Prox solvers.
>
> R3's description of the forward pass through the network with a Prox solver is correct. In general, the best way to understand how a network with a Prox solver operates in both forward and backward passes is to think of that layer with a Prox solver as a recurrent neural network. Thus, asking how one performs a backward pass through such a layer is equivalent to asking how one would perform a backward pass through a recurrent neural network. The backward pass through the Prox solver is still performed: not through the original network as R3 thought, but through the same network with the Prox solver, akin to backpropagation-through-time (BPTT). This resembles the parameter update procedure in recurrent neural networks.
>
> (2) Adjusting baselines to perform the same amount of computation for a fair comparison.
>
> R3 is correct about the fact that networks with Prox solvers do in fact perform more computation. However, the capacity of both networks (baseline and the Prox solver network) is identical. This means that both networks have the same exact number of parameters and there is no advantage of the Prox solvers in terms of capacity over the baseline. This is the essential factor for a fair comparison, since accuracy is often considered as a function of network capacity rather than the amount of computation. Moreover, note that to the best of our knowledge we have reported the best results for the baselines in comparison to any publicly available online repository that performs training without using any test data statistics (i.e. proper training). For instance, the results of VGG16 on CIFAR-10 are comparable to or better than some ResNet architectures on the same dataset. We are not aware of better numbers for the corresponding networks on the corresponding datasets.
>
> On the minor issues.
>
> (1) Missing definitions.
>
> We have addressed this in the revised version. We provided several examples of $g(\mathbf{x})$ when it was first introduced in the first page. We have explicitly, as suggested, defined $p$ in Proposition 1.
>
> (2) Give examples of where the prox problems in Table 1 show up in practice (outside of activation functions in neural networks).
>
> We are only aware of applications in activation functions in neural networks. But this is sufficient motivation for us.
>
> (3) On the statement ``for different choices of dropout rate the baseline can always be improved by..." in the Experiments.
>
> We have softened the statement in the revision. The statement is now ``we observe that for different choices of dropout rate the baseline performance improves upon replacing ...".
>
> (4) Include results for Dropout rate p=0 in Table 5.
>
> We have added this experiment to Table 5.

---

### Public Comment · (anonymous) · 2018-10-09
**Relation to incremental proximal gradient methods**

The authors provide an interesting view on layers of typical feed forward (C)NNs by drawing connections to proximal operators and thus convex optimization. In Section 3.2 equation (3), the authors highlight that a single layer can be interpreted as a proximal gradient step on a convex function F(x) + g(x). A closely related connection was also drawn by [1] for residual networks. Kobler et al. analyzed the connections to incremental proximal gradient methods for both convex and non-convex F(x). Using this relation, Kobler et al. interpreted a block of residual layers as a sequence of proximal incremental gradient steps that minimize composite functions just like equation (1) in this paper.

Due to the aforementioned relations to [1], the authors should consider citing [1].

[1] Kobler et al. "Variational networks: connecting variational methods and deep learning", German Conference on Pattern Recognition, 2017.

---

> ### Author Response · Authors · 2018-10-10
> **Thank you for the feedback**
>
> Thank you for pointing out this reference which is indeed relevant to our work. We will include it in the revised version of the paper. Note that Kobler et al. propose an architecture (variational networks) for the task of image reconstruction that is motivated by approximate connections between ResNet blocks and proximal point methods. Their approach naturally fits in the line of work discussed in our Related Work section on the merits of guiding the design of deep networks using optimization algorithms to derive architectures for a given task (such as image reconstruction). In our work, we study the connections between general, existing models and optimization problems. We explain Dropout layers in this framework and introduce a practical way to compute the Lipschitz constant. This allows us to introduce the general strategy of replacing layers of existing architectures with solvers.

---

### Author Response · Authors · 2018-11-13
**Comment on a concurrent ICLR submission**

We would like to bring to the attention of the reviewers a well-rated concurrent ICLR submission, ``"The Singular Values of Convolutional Layers". The main result of that paper (Theorem 6) is equivalent to Lemma 2 in our submission. In our submission, this is a supporting result, the proof of which is relegated to the supplement. Our proof is simple and is one page long. This is enabled by Lemma 1, which enables the use of tensor theory in connecting convolutional and fully-connected layers in a unified framework. We believe this speaks to the value of our work as a whole.

---

### Meta-Review · Area_Chair1 · 2018-12-13
**Interesting new perspective on deep learning**

**Confidence:** 5
**Recommendation:** Accept (Poster)

**Metareview:**

This paper relates deep learning to convex optimization by showing that the forward pass though a dropout layer, linear layer (either convolutional or fully connected), and a nonlinear activation function is equivalent to taking one τ-nice proximal gradient descent step on a a convex optimization objective. The paper shows (1) how different activation functions correspond to different proximal operators, (2) that replacing Bernoulli dropout with additive dropout corresponds to replacing the τ-nice proximal gradient descent method with a variance-reduced proximal method, and (3) how to compute the Lipschitz constant required to set the optimal step size in the proximal step. The practical value of this perspective is illustrated in experiments that replace various layers in ConvNet architectures with proximal solvers, leading to performance improvements on CIFAR-10 and CIFAR-100. The reviewers felt that most of their concerns were adequately addressed in the discussion and revision, and that the paper should be accepted.